# RANDOM MATRICES IN SERVICE OF ML FOOTPRINT: TERNARY RANDOM FEATURES WITH NO PERFORMANCE LOSS

**Hafiz Tiomoko Ali**
Huawei Noah's Ark Lab (London)
hafiz.tiomoko.ali@huawei.com

**Zhenyu Liao**
Huazhong University of Science & Technology, China
zhenyu_liao@hust.edu.cn

**Romain Couillet**
University Grenoble Alpes, Inria, CNRS, Grenoble INP, LIG, 38000 Grenoble, France
romain.couillet@univ-grenoble-alpes.fr

## ABSTRACT

In this article, we investigate the spectral behavior of random features kernel matrices of the type $\mathbf{K} = \mathbb{E}_{\mathbf{w}}[\sigma(\mathbf{w}^\mathsf{T}\mathbf{x}_i)\sigma(\mathbf{w}^\mathsf{T}\mathbf{x}_j)]_{i,j=1}^n$, with nonlinear function $\sigma(\cdot)$, data $\mathbf{x}_1, \ldots, \mathbf{x}_n \in \mathbb{R}^p$, and random projection vector $\mathbf{w} \in \mathbb{R}^p$ having i.i.d. entries. In a high-dimensional setting where the number of data $n$ and their dimension $p$ are both large and comparable, we show, under a Gaussian mixture model for the data, that the eigenspectrum of $\mathbf{K}$ is *independent* of the distribution of the i.i.d. (zero-mean and unit-variance) entries of $\mathbf{w}$, and *only* depends on $\sigma(\cdot)$ via its (generalized) Gaussian moments $\mathbb{E}_{z\sim\mathcal{N}(0,1)}[\sigma'(z)]$ and $\mathbb{E}_{z\sim\mathcal{N}(0,1)}[\sigma''(z)]$. As a result, for any kernel matrix $\mathbf{K}$ of the form above, we propose a novel random features technique, called Ternary Random Feature (TRF), that (i) asymptotically yields the same limiting kernel as the original $\mathbf{K}$ in a spectral sense and (ii) can be computed and stored much more efficiently, by wisely tuning (in a *data-dependent* manner) the function $\sigma$ and the random vector $\mathbf{w}$, both taking values in $\{-1, 0, 1\}$. The computation of the proposed random features requires no multiplication, and a factor of $b$ times less bits for storage compared to classical random features such as random Fourier features, with $b$ the number of bits to store full precision values. Besides, it appears in our experiments on real data that the substantial gains in computation and storage are accompanied with somewhat improved performances compared to state-of-the-art random features compression/quantization methods.

## 1 INTRODUCTION

Kernel methods are among the most powerful machine learning approaches with a wide range of successful applications (Schölkopf & Smola, 2018) which, however, suffer from scalability issues in large-scale problems, due to their high space and time complexities (with respect to the number of data $n$). To address this key limitation, a myriad of random features based kernel approximation techniques have been proposed (Rahimi & Recht, 2008; Liu et al., 2021a): random features methods randomly project the data to obtain low-dimensional nonlinear representations that approximate the original kernel features. This allows practitioners to apply them, with a large saving in both time and space, to various kernel-based downstream tasks such as kernel spectral clustering (Von Luxburg, 2007), kernel principal component analysis (Schölkopf et al., 1997), kernel canonical correlation analysis (Lai & Fyfe, 2000), kernel ridge regression (Vovk, 2013), to name a few. A wide variety of these kernels can be written, for data points $\mathbf{x}_i, \mathbf{x}_j \in \mathbb{R}^p$, in the form

$$\kappa(\mathbf{x}_i, \mathbf{x}_j) = \mathbb{E}_{\mathbf{w}}\left[\sigma\left(\mathbf{w}^\mathsf{T}\mathbf{x}_i\right)\sigma\left(\mathbf{w}^\mathsf{T}\mathbf{x}_j\right)\right] \tag{1}$$

with $\mathbf{w} \in \mathbb{R}^p$ having i.i.d. entries, which can be "well approximated" by a sample mean $\frac{1}{m}\sum_{t=1}^m \sigma\left(\mathbf{w}_t^\mathsf{T}\mathbf{x}_i\right)\sigma\left(\mathbf{w}_t^\mathsf{T}\mathbf{x}_j\right)$ over $m$ independent random features for $m$ sufficiently large. For instance, taking $\sigma(x) = [\cos(x), \sin(x)]$ and $\mathbf{w}$ with i.i.d. standard Gaussian entries, one obtains the popular Random Fourier Features (RFFs) that approximate the Gaussian kernel (and the

Laplacian kernel for Cauchy distributed $\mathbf{w}$ with the same choice of $\sigma$) (Rahimi & Recht, 2008); for $\sigma(x) = \max(x, 0)$, one approximates the first order Arc-cosine kernel; and the zeroth order Arc-cosine kernel (Cho, 2012) with $\sigma(x) = (1 + \text{sign}(x))/2$, etc.

As shall be seen subsequently, (random) neural networks are, to a large extent, connected to *kernel matrices* of the form (1). More specifically, the classification or regression performance at the output of random neural networks are functionals of random matrices that fall into the wide class of kernel random matrices. Perhaps more surprisingly, this connection still exists for *deep neural networks* which are (i) randomly initialized and (ii) trained with gradient descent, as testified by the recent works on *neural tangent kernels* (Jacot et al., 2018), by considering the "infinitely many neurons" limit, that is, the limit where the network widths of all layers go to infinity simultaneously. This close connection between neural networks and kernels has triggered a renewed interest for the theoretical investigation of deep neural networks from various perspectives, including optimization (Du et al., 2019; Chizat et al., 2019), generalization (Allen-Zhu et al., 2018; Arora et al., 2019; Bietti & Mairal, 2019), and learning dynamics (Lee et al., 2019; Advani et al., 2020; Liao & Couillet, 2018a). These works shed new light on the theoretical understanding of deep neural network models and specifically demonstrate the significance of studying networks with random weights and their associated kernels to assess the mechanisms underlying more elaborate deep networks.

In this article, we consider the random feature kernel of the type (1), which can also be seen as the limiting kernel of a single-hidden-layer neural network with a random first layer. By assuming a high-dimensional Gaussian Mixture Model (GMM) for the data $\{\mathbf{x}_i\}_{i=1}^n$ with $\mathbf{x}_i \in \mathbb{R}^p$, we show that the *centered* kernel matrix[1]

$$\mathbf{K} \triangleq \mathbf{P}\{\kappa(\mathbf{x}_i, \mathbf{x}_j)\}_{i,j=1}^n \mathbf{P}, \quad \mathbf{P} \triangleq \mathbf{I}_n - \frac{1}{n}\mathbf{1}_n\mathbf{1}_n^\mathsf{T}, \tag{2}$$

is asymptotically (as $n, p \to \infty$ with $p/n \to c \in (0, \infty)$) equivalent, in a spectral sense, to another random kernel matrix $\tilde{\mathbf{K}}$ which depends on the GMM data statistics and the generalized Gaussian moments $\mathbb{E}[\sigma'(z)], \mathbb{E}[\sigma''(z)]$ of the activation function $\sigma(\cdot)$, but is *independent* of the specific law of the i.i.d. entries of the random vector $\mathbf{w}$, as long as they are normalized to have zero mean and unit variance. As such, one can design novel random features schemes with limiting kernels asymptotically equivalent to the original $\mathbf{K}$. For instance, define

$$\kappa^{ter}(\mathbf{x}_i, \mathbf{x}_j) \triangleq \mathbb{E}_{\mathbf{w}^{ter}}\left[\sigma^{ter}\left(\mathbf{x}_i^\mathsf{T}\mathbf{w}^{ter}\right)\sigma^{ter}\left(\mathbf{x}_j^\mathsf{T}\mathbf{w}^{ter}\right)\right] \tag{3}$$

with $\mathbf{w}^{ter} \in \mathbb{R}^p$ having i.i.d. entries taking value $w_i^{ter} = 0$ (with probability $\epsilon$) and value

$$w_i^{ter} \in \left\{-(1-\epsilon)^{-\frac{1}{2}}, (1-\epsilon)^{-\frac{1}{2}}\right\} \tag{4}$$

each with probability $1/2 - \epsilon/2$, where $\epsilon \in [0, 1)$ represents the *level of sparsity* of $\mathbf{w}$, and

$$\sigma^{ter}(t) = -1 \cdot \delta_{t<s_-} + 1 \cdot \delta_{t>s_+} \tag{5}$$

for some thresholds $s_- < s_+$ chosen to match the generalized Gaussian moments $\mathbb{E}[\sigma'(z)], \mathbb{E}[\sigma''(z)]$ of *any* $\sigma$ function (e.g., ReLU, $\cos$, $\sin$) widely used in random features or neural network contexts. The proposed Ternary Random Features (TRFs, with limiting kernel matrices defined in (3) asymptotically "matching" *any* random features kernel matrices in a spectral sense) have the computational advantage of being sparse and not requiring multiplications but only additions, as well as the storage advantage of being composed of only a finite set of words, e.g., $\{-1, 0, 1\}$ for $\epsilon = 0$.

Given the urgent need for environmentally-friendly, but still efficient, neural networks such as binary neural networks (Hubara et al., 2016; Lin et al., 2015; Zhu et al., 2016; Qin et al., 2020; Hubara et al., 2016), pruned neural networks (Liu et al., 2015; Han et al., 2015a;b), weights-quantized neural networks (Gupta et al., 2015; Gong et al., 2014), we hope that our analysis will open a new door to a *random matrix-improved* framework of computationally efficient methods for machine learning and neural network models more generally.

---

[1]Left- and right-multiplying the kernel matrices by $\mathbf{P}$ is equivalent to centering the data in the kernel feature space, which is a common practice in kernel learning and plays a crucial role in multidimensional scaling (Joseph & Myron, 1978) and kernel PCA (Schlköpf et al., 1998). In the remainder of this paper, whenever we use kernel matrices, they are considered to have been centered in this way.

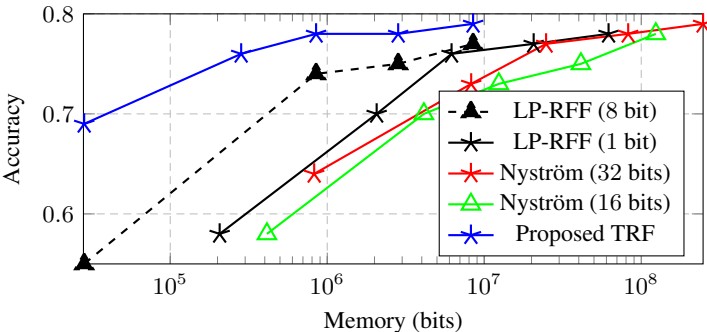

Figure 1: Test accuracy of logistic regression on quantized random features for different number of features $m \in \{10^2, 10^3, 5.10^3, 10^4, 5.10^4\}$, with LP-RFF (8-bit and 1-bit, in **black**) (Zhang et al., 2019), Nyström approximation (32 bits in **red**, 16 bits in **green**) (Williams & Seeger, 2001), versus the proposed TRF approach (in **blue**), on the two-class Cov-Type dataset from UCI ML repo, with $n = 418\,000$ training samples, $n_{test} = 116\,000$ test samples, and data dimension $p = 54$.

## 1.1 CONTRIBUTIONS

Our main results are summarized as follows.

1. By considering a high-dimensional Gaussian mixture model for the data, we show (Theorem 1) that for $\mathbf{K}$ defined in (2), $\|\mathbf{K} - \tilde{\mathbf{K}}\| \to 0$ as $n, p \to \infty$, where $\tilde{\mathbf{K}}$ is a random matrix *independent* of the law of $\mathbf{w}$, and depends on the nonlinear $\sigma(\cdot)$ *only* via its generalized Gaussian moments $\mathbb{E}[\sigma'(z)]$ and $\mathbb{E}[\sigma''(z)]$ for $z \sim \mathcal{N}(0, 1)$.

2. We exploit this result to propose a computationally efficient random features approach, called Ternary Random Features (TRFs), with asymptotically the same limiting kernel as *any* random features-type kernel matrices $\mathbf{K}$ of the form (2), while inducing no multiplication and $b$ times less memory storage for its computation, with $b$ the number of bits to store full precision values, e.g., $b = 32$ in the case of a single-precision floating-point format.

3. We provide empirical evidence on various random-features based algorithms, showing the computational and storage advantages of the proposed TRF method, while achieving competitive performances compared to state-of-the-art random features techniques. As a first telling example, in Figure 1 on the Cov-Type dataset from the UCI ML repository, TRFs achieve similar logistic regression performance as the LP-RFF method proposed by Zhang et al. (2019), with 8 times less memory, and 32 or 16 times less memory than the Nyström approximation of the Gaussian kernel (using full precision of 32 or 16 bits) (Williams & Seeger, 2001); see the shift in the x-axis (memory) of Figure 1.

## 1.2 RELATED WORK

**Random features kernel and random neural networks.** Random features methods were first proposed to relieve the computational and storage burden of kernel methods in large-scale problems when the number of training samples $n$ is large (Schölkopf & Smola, 2018; Rahimi & Recht, 2008; Liu et al., 2021a). For instance, Random Fourier Features can be used to approximate the popular Gaussian kernel, when the number of random features $m$ is sufficiently large (Rahimi & Recht, 2008). Since (deep) modern neural networks are routinely trained with random initialization, random features method is also considered a stylish model to analyze neural networks (Neal, 1996; Williams, 1997; Novak et al., 2018; Matthews et al., 2018; Lee et al., 2017; Louart et al., 2018). In particular, by focusing on the large $n, m$ regime, the analysis of such models led to the so-called *double descent* theory (Advani et al., 2020; Mei & Montanari, 2019; Liao et al., 2020) for neural nets.

**Computationally efficient random features methods.** In an effort to further reduce the computation and storage costs of random features models, various quantization and binarization methods were proposed (Goemans & Williamson, 1995; Charikar, 2002; Li & Slawski, 2017; Li & Li, 2019; 2021;

Agrawal et al., 2019; Zhang et al., 2019; Liao et al., 2021; Couillet et al., 2021). More precisely, Agrawal et al. (2019) combined RFFs with a data-dependent feature selection approach to reduce the computational cost, while preserving the statistical guarantees of (using the original set of) RFFs. Zhang et al. (2019) proposed a low-precision approximation of RFFs to significantly reduce the storage while generalizing as well as full-precision RFFs. Li & Li (2021) designed quantization schemes of RFFs for arbitrary choice of the Gaussian kernel parameter. Our work improves these previous efforts by (i) considering a broader family of random features kernels beyond RFFs and by (ii) proposing the TRF approach that is both sparse and quantized, while asymptotically yielding the *same* limiting kernel spectral structure, and thus algorithmic performances (Cortes et al.).

**Random matrix theory and neural networks.** Random matrix theory (RMT), as a powerful and flexible tool to investigate the (asymptotic) behavior of large-scale systems, is recently gaining popularity in the analysis of (deep) neural networks (Martin & Mahoney, 2018; 2019). In this respect, Pennington & Worah (2017) derived the eigenvalue distribution of the Conjugate Kernel (CK) in a single-hidden-layer random neural network model. This result was then generalized to a broader class of data distributions (Louart et al., 2018) and to a multi-layer scenario (Benigni & Péché, 2019; Pastur, 2020). Fan & Wang (2020) went beyond the general i.i.d. assumption (on the entries of data vectors) and studied the spectral properties of the CK and neural tangent kernel for data that are approximately "pairwise orthogonal." Our work improves (Fan & Wang, 2020) by studying the random features kernel for more structured GMM data, and is thus more adapted to machine learning applications such as classification. As far as the study of random features kernels under GMM data is concerned, the closest work to ours is (Liao & Couillet, 2018b) where the kernel matrix $\mathbf{K}$ defined in (2) is studied for GMM data, but only for a few specific activation functions and Gaussian $\mathbf{w}$, see Footnote 5 below for a detailed account of the technical differences between this work and (Liao & Couillet, 2018b). Here, we provide a *universal* result with respect to the (much broader class of) activation functions and random $\mathbf{w}$, and propose a computation and storage efficient random features technique well tuned to match the performances of *any* commonly used random features kernels.

## 1.3 NOTATIONS AND ORGANIZATION OF THE ARTICLE

In this article, we denote scalars by lowercase letters, vectors by bold lowercase, and matrices by bold uppercase. We denote the transpose operator by $(\cdot)^{\mathsf{T}}$, we use $\|\cdot\|$ to denote the Euclidean norm for vectors and spectral/operator norm for matrices. For a random variable $z$, $\mathbb{E}[z]$ denotes the expectation of $z$. The notation $\delta_{x \in A}$ is the Kronecker delta taking value 1 when $x \in A$ and 0 otherwise. Finally, $\mathbf{1}_p$ and $\mathbf{I}_p$ are respectively the vector of all one's of dimension $p$ and the identity matrix of dimension $p \times p$.

The remainder of the article is structured as follows. In Section 2, we describe the random features model under study along with our working assumptions. We then present our main technical results in Section 3 on the spectral characterization of random features kernel matrices $\mathbf{K}$ and its practical consequences, in particular the design of cost-efficient ternary random features (TRFs) leading asymptotically to the same kernel as any generic random features. We provide empirical evidence in Section 4 showing the computational and storage advantage along with competitive performance of TRFs compared to state-of-the-art random features approaches. Conclusion and perspective are placed in Section 5.

## 2 SYSTEM SETTINGS

Let $\mathbf{W} \in \mathbb{R}^{m \times p}$ be a random matrix having i.i.d. entries with zero mean and unit variance. The random features matrix $\mathbf{\Sigma} \in \mathbb{R}^{m \times n}$ of some data $\mathbf{X} \in \mathbb{R}^{p \times n}$ is defined as $\mathbf{\Sigma} \triangleq \sigma(\mathbf{WX})$ for some nonlinear activation function $\sigma : \mathbb{R} \to \mathbb{R}$ applied entry-wise on $\mathbf{WX}$. We denote $\mathbf{G}$ the associated random features Gram matrix

$$\mathbf{G} \triangleq \frac{1}{m}\mathbf{\Sigma}^{\mathsf{T}}\mathbf{\Sigma} = \frac{1}{m}\sigma(\mathbf{WX})^{\mathsf{T}}\sigma(\mathbf{WX}) = \frac{1}{m}\sum_{t=1}^{m}\sigma\left(\mathbf{X}^{\mathsf{T}}\mathbf{w}_t\right)\sigma\left(\mathbf{w}_t^{\mathsf{T}}\mathbf{X}\right) \in \mathbb{R}^{n \times n} \quad (6)$$

which is a sample mean of the *expected* kernel defined in (1). With $\mathbf{P} \triangleq \mathbf{I}_n - \frac{1}{n}\mathbf{1}_n\mathbf{1}_n^{\mathsf{T}}$, we consider the *expected* and *centered* random features kernel matrix $\mathbf{K}$ defined in (2), which plays a fundamental

role in various random features kernel-based learning methods such as kernel ridge regression, logistic regression, support vector machines, principal component analysis, or spectral clustering.

Let $\mathbf{x}_1, \cdots, \mathbf{x}_n \in \mathbb{R}^p$ be $n$ independent data vectors belonging to one of $K$ distributional classes $\mathcal{C}_1, \cdots, \mathcal{C}_K$, with class $\mathcal{C}_a$ having cardinality $n_a$. We assume that $\mathbf{x}_i$ follows a Gaussian Mixture Model (GMM), that is, for $\mathbf{x}_i \in \mathcal{C}_a$,

$$\mathbf{x}_i = \boldsymbol{\mu}_a/\sqrt{p} + \mathbf{z}_i \tag{7}$$

with $\mathbf{z}_i \sim \mathcal{N}(\mathbf{0}_p, \mathbf{C}_a/p)$ for some mean $\boldsymbol{\mu}_a \in \mathbb{R}^p$ and covariance $\mathbf{C}_a \in \mathbb{R}^{p \times p}$ associated to class $\mathcal{C}_a$.

In high dimensions, under "non triviality assumptions",[2] the data vectors drawn from (7) can be shown to be neither very "close" nor very "far" from each other, irrespective of the class they belong to, see (Couillet et al., 2016). We place ourselves under such non-trivial conditions, by imposing, as in (Couillet et al., 2016; 2018), the following growth rate conditions.

**Assumption 1 (High-dimensional asymptotics)** *As $n \to \infty$, we have (i) $p/n \to c \in (0, \infty)$ and $n_a/n \to c_a \in (0, 1)$; (ii) $\|\boldsymbol{\mu}_a\| = O(1)$; (iii) for $\mathbf{C}^\circ = \sum_{a=1}^K \frac{n_a}{n} \mathbf{C}_a$ and $\mathbf{C}_a^\circ = \mathbf{C}_a - \mathbf{C}^\circ$, $\|\mathbf{C}_a\| = O(1)$, $\mathrm{tr}(\mathbf{C}_a^\circ) = O(\sqrt{p})$ and $\mathrm{tr}(\mathbf{C}_a \mathbf{C}_b) = O(p)$ for $a, b \in \{1, \cdots, K\}$. We denote $\tau \triangleq \mathrm{tr}(\mathbf{C}^\circ)/p$ that is assumed to converge in $(0, \infty)$.*

**Remark 1 (Beyond Gaussian mixture data)** *While the theoretical results in this paper are derived for Gaussian mixture data in (7), under the non-trivial setting of Assumption 1, we conjecture that they can be extended to a much broader family of data distributions beyond the Gaussian setting, e.g., to the so-called* concentrated random vector *family (Seddik et al., 2020; Louart & Couillet, 2018), under similar non triviality assumptions. See Section A.1 in the appendix for more discussions.*

To cover a large family of random features, we assume that the random projector matrix $\mathbf{W}$ has i.i.d. entries of zero mean, unit variance and bounded fourth-order moment, with no restriction on their particular distribution.

**Assumption 2 (On random projection matrix)** *The random matrix $\mathbf{W}$ has i.i.d. entries such that $\mathbb{E}[W_{ij}] = 0$, $\mathbb{E}[W_{ij}^2] = 1$ and $\mathbb{E}[W_{ij}^4] < \lambda_W$ for some constant $\lambda_W < \infty$.*

We consider the family of activation functions $\sigma(\cdot)$ satisfying the following assumption.

**Assumption 3 (On activation function)** *The function $\sigma$ is at least twice differentiable (in the sense of distributions when applied to a random variable having a non-degenerate distribution function, see Remark 3 in Section A.1 of the appendix), with $\max\{\mathbb{E}|\sigma(z)|, \mathbb{E}|\sigma^2(z)|, \mathbb{E}|\sigma'(z)|, \mathbb{E}|\sigma''(z)|\} < \lambda_\sigma$ for some constant $\lambda_\sigma < \infty$ and $z \sim \mathcal{N}(0, 1)$.*

## 3  MAIN RESULT

Our objective is to characterize the high-dimensional spectral behavior of the centered and expected random features kernel matrix $\mathbf{K}$ defined in (2). It turns out, somewhat surprisingly, that under the non-trivial setting of Assumption 1, one may mentally picture the high-dimensional data vectors as (i) being asymptotically pairwise *orthogonal* (i.e., $\mathbf{x}_i^\mathsf{T} \mathbf{x}_j \to 0$ for $i \neq j$ as $p \to \infty$) and (ii) having asymptotically *equal* Euclidean norms (i.e., $\|\mathbf{x}_i\|^2 \to \tau$), *independently* of the underlying class they belong to. As we shall see, this high-dimensional "concentration" of $\mathbf{x}_i^\mathsf{T} \mathbf{x}_j \to \tau \cdot \delta_{i=j}$ plays a crucial role in "decoupling" the two *dependent* but asymptotically Gaussian random variables $\sigma(\mathbf{w}^\mathsf{T} \mathbf{x}_i)$ and $\sigma(\mathbf{w}^\mathsf{T} \mathbf{x}_j)$ in the definition (1). This, up to some careful control on the higher-order (but well "concentrated") terms, leads to the following result on the asymptotic behavior of $\mathbf{K}$, the proof of which is given in Section A.3 of the appendix.

**Theorem 1 (Asymptotic equivalent of K)** *Under Assumption 1-3, for $\mathbf{K}$ defined in (2), as $n \to \infty$,*

$$\|\mathbf{K} - \tilde{\mathbf{K}}\| \to 0,$$

---

[2]That is, when classification is neither too hard nor too easy.

*almost surely with* $\mathbf{P} = \mathbf{I}_n - \mathbf{1}_n \mathbf{1}_n^\mathsf{T}/n$,

$$\tilde{\mathbf{K}} = \mathbf{P}\left( d_1 \cdot \left( \mathbf{Z} + \mathbf{M}\frac{\mathbf{J}^\mathsf{T}}{\sqrt{p}} \right)^\mathsf{T} \left( \mathbf{Z} + \mathbf{M}\frac{\mathbf{J}^\mathsf{T}}{\sqrt{p}} \right) + d_2 \cdot \mathbf{V}\mathbf{A}\mathbf{V}^\mathsf{T} + d_0 \cdot \mathbf{I}_n \right)\mathbf{P}, \qquad (8)$$

*and*

$$\mathbf{V} = [\mathbf{J}/\sqrt{p}, \; \boldsymbol{\phi}] \in \mathbb{R}^{n \times (K+1)}, \quad \mathbf{A} = \begin{bmatrix} \mathbf{t}\mathbf{t}^\mathsf{T} + 2\mathbf{T} & \mathbf{t} \\ \mathbf{t}^\mathsf{T} & 1 \end{bmatrix} \in \mathbb{R}^{(K+1) \times (K+1)},$$

*where, for* $z \sim \mathcal{N}(0,1)$,

$$d_0 = \mathbb{E}[\sigma^2(\sqrt{\tau}z)] - \mathbb{E}[\sigma(\sqrt{\tau}z)]^2 - \tau\mathbb{E}[\sigma'(\sqrt{\tau}z)]^2, \quad d_1 = \mathbb{E}[\sigma'(\sqrt{\tau}z)]^2, \quad d_2 = \frac{1}{4}\mathbb{E}[\sigma''(\sqrt{\tau}z)]^2$$

*and "first-order" random matrix* $\mathbf{Z} = [\mathbf{z}_1, \cdots, \mathbf{z}_n] \in \mathbb{R}^{p \times n}$ *as defined in (7), "second-order" random (fluctuation) vector* $\boldsymbol{\phi} = \left\{ \|\mathbf{z}_i\|^2 - \mathbb{E}[\|\mathbf{z}_i\|^2] \right\}_{i=1}^n \in \mathbb{R}^n$, *and GMM data statistics*

$$\mathbf{M} = [\boldsymbol{\mu}_1, \cdots, \boldsymbol{\mu}_K] \in \mathbb{R}^{p \times K}, \; \mathbf{t} = \left\{ \frac{\mathrm{tr}(\mathbf{C}_a^\circ)}{\sqrt{p}} \right\}_{a=1}^K \in \mathbb{R}^K, \; \mathbf{T} = \left\{ \frac{\mathrm{tr}\mathbf{C}_a\mathbf{C}_b}{p} \right\}_{a,b=1}^K \in \mathbb{R}^{K \times K} \quad (9)$$

*as well as the class label vectors* $\mathbf{J} = [\mathbf{j}_1, \cdots, \mathbf{j}_K] \in \mathbb{R}^{n \times K}$ *with* $[\mathbf{j}_a]_i = \delta_{\mathbf{x}_i \in \mathcal{C}_a}$.

First note that the "asymptotic equivalence" established in Theorem 1 holds for the *expected* kernel $\mathbf{K}$, not on the *empirical* random feature kernel $\mathbf{G}$ defined in (6), and is thus *independent* of the number of random features $m$. On closer inspection of Theorem 1, we see that the data statistics for classification, i.e., the means ($\mathbf{M}$) and covariances ($\mathbf{t}\mathbf{t}^\mathsf{T}$, $\mathbf{T}$) are respectively weighted by the generalized Gaussian moments of first ($d_1$) and second order ($d_2$), while the coefficient $d_0$ merely acts as a regularization term to shift *all* the eigenvalues,[3] and has thus asymptotically *no* impact on the performance of, e.g., kernel spectral clustering for which only eigenvector structures are exploited.[4]

Theorem 1 unveils a surprising *universal* behavior of the expected kernel $\mathbf{K}$ in the large $n, p$ regime. Specifically, the expression of $\tilde{\mathbf{K}}$ (and thus the spectral behavior of $\mathbf{K}$, see our discussion in the paragraph that follows) is *universal* with respect to the distribution of $\mathbf{W}$ and the activation function $\sigma$, when they are "normalized" to satisfy Assumptions 2 and 3. This technically improves previous efforts such as (Liao & Couillet, 2018b),[5] and allows us to design computationally more efficient random features approach by wisely choosing $\mathbf{W}$ and $\sigma$.

As a direct consequence of Theorem 1, one has, by Weyl's inequality and Davis–Kahan theorem that (i) the difference between each corresponding pair of eigenvalues and (ii) the distance between the "isolated" eigenvectors or the "angle" between the "isolated subspaces" of $\mathbf{K}$ and $\tilde{\mathbf{K}}$ vanish asymptotically as $n, p \to \infty$. This is numerically confirmed in Figure 2, where one observes a close match between the spectra (eigenvalue "bulk" and isolated eigenvalue-eigenvector pairs) of $\mathbf{K}$ and those of its asymptotic equivalent $\tilde{\mathbf{K}}$ given in Theorem 1, already for $n, p$ only in hundreds. In particular, we compare, in Figure 2, the eigenspectra of $\mathbf{K}$ and $\tilde{\mathbf{K}}$ for Gaussian versus Student-t distributed $\mathbf{W}$, on GMM data. A close match of the spectra and the isolated eigen-pairs is observed, irrespective of the distribution of $\mathbf{W}$, as predicted by our theory.

In the following corollary, we exploit the universal result in Theorem 1 to design computationally efficient Ternary Random Features (TRFs) by specifying the distribution of (the entries of) $\mathbf{W}$ to symmetric Bernoulli and $\sigma$ to a ternary function, so as to obtain a limiting kernel $\mathbf{K}^{ter}$ that is asymptotically equivalent to *any* random features kernel matrix $\mathbf{K}$ of the form (2), for the high-dimensional GMM data under study. This is proved in Section A.4 of the appendix.

---

[3]This can be seen as another manifestation of the *implicit regularization* in high-dimensional kernel and random features (Jacot et al., 2020; Derezinski et al., 2020; Liu et al., 2021b).

[4]We provide in Table 1 (Section A.5 of the appendix) the corresponding Gaussian moments $d_0, d_1, d_2$ for various commonly used activation functions in random features and neural network contexts.

[5] From a technical perspective, Theorem 1 improves (Liao & Couillet, 2018b, Theorem 1) in the fact that the latter relies on the *explicit* forms of the expectation $\mathbf{K}$ in (2), and is thus limited to (i) a few nonlinear $\sigma$ for which $\mathbf{K}$ can be computed explicitly, see (Liao & Couillet, 2018b, Table 1); and (ii) Gaussian distributed $\mathbf{W}$ in which case the $p$-dimensional integral can be easily reduced to a two-dimensional one. Here, Theorem 1 holds for a much broader family of random $\mathbf{W}$ and nonlinear $\sigma$ as long as Assumptions 2 and 3 hold.

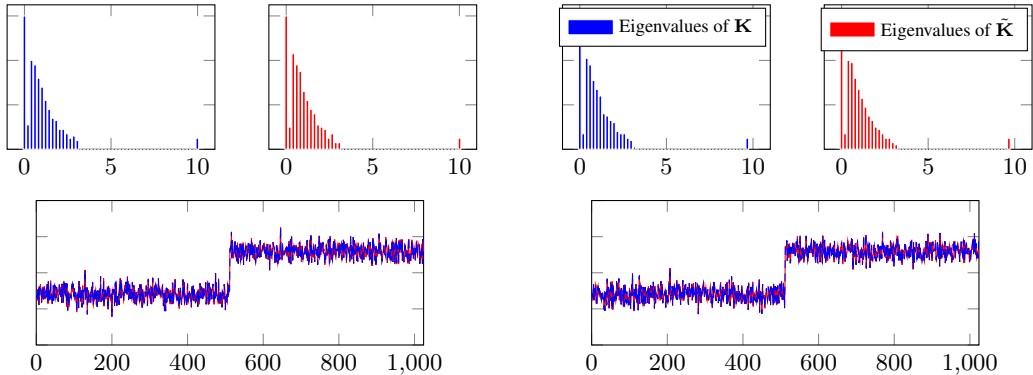

Figure 2: Eigenvalue distribution (**TOP**) and eigenvector associated to the largest eigenvalue (**BOTTOM**) of the expected and centered kernel matrix $\mathbf{K}$ (**blue**) versus its asymptotic equivalent $\tilde{\mathbf{K}}$ (**red**) in Theorem 1, with $\sigma(t) = \max(t, 0)$. (**LEFT**) $\mathbf{W}$ having **Gaussian** entries and (**RIGHT**) $\mathbf{W}$ having **Student-t** entries with 7 degrees of freedom, for two-class GMM data with $\boldsymbol{\mu}_a = [\mathbf{0}_{a-1}; 4; \mathbf{0}_{p-a}], \mathbf{C}_a = (1 + 4(a-1)/\sqrt{p})\mathbf{I}_p$, $p = 512$ and $n = 2048$.

**Corollary 1 (Ternary Random Features)** *For a given random features kernel matrix $\mathbf{K}$ of the form (2) with $\mathbf{W}$ and nonlinear $\sigma$ satisfying Assumptions 1-3, with associated generalized Gaussian moments $d_0, d_1, d_2$ defined in Theorem 1, let $\sigma^{ter}$ be defined in (5) with $s_- = \hat{s}_-$, $s_+ = \hat{s}_+$, and $\hat{s}_-$, $\hat{s}_+$ satisfying the following equations*

$$d_1 = \frac{1}{\pi^2}\left(e^{-\hat{s}_+^2/\tau} + e^{-\hat{s}_-^2/\tau}\right)^2, \quad d_2 = \frac{1}{2\pi\tau^3}\left(\hat{s}_+ e^{-\hat{s}_+^2/\tau} + \hat{s}_- e^{-\hat{s}_-^2/\tau}\right)^2. \tag{10}$$

*Define the Ternary Random Features matrix $\boldsymbol{\Sigma}^{ter} = \sigma^{ter}(\mathbf{W}^{ter}\mathbf{X})$ with $\mathbf{W}^{ter}$ defined in (4) having sparsity level $\epsilon$, the associated Gram matrix $\mathbf{G}^{ter} = \frac{1}{m}(\boldsymbol{\Sigma}^{ter})^\mathsf{T}\boldsymbol{\Sigma}^{ter}$ as in (6), and the limiting kernel*

$$\mathbf{K}^{ter} \triangleq \mathbf{P}\{\kappa^{ter}(\mathbf{x}_i, \mathbf{x}_j)\}_{i,j=1}^n \mathbf{P} \tag{11}$$

*for $\kappa^{ter}$ defined in (3). Then, there exists $\lambda \in \mathbb{R}$ such that*[6] $\|\mathbf{K} - \mathbf{K}^{ter} - \lambda\mathbf{P}\| \to 0$ *almost surely as $n \to \infty$.*

Note that the system of equations in (10) defining $\hat{s}_-$ and $\hat{s}_+$ is a function of the key parameter $\tau = \operatorname{tr}\mathbf{C}^\circ/p$ defined in Assumption 1, which can be consistently estimated from the data; see Algorithm 1 below and a proof in Lemma 1 of the appendix (the intuition of which follows from the fact that $\|\mathbf{x}_i\|^2 = \|\boldsymbol{\mu}_a\|^2/p + 2\boldsymbol{\mu}_a^\mathsf{T}\mathbf{z}_i/\sqrt{p} + \|\mathbf{z}_i\|^2 = \mathbb{E}[\operatorname{tr}(\mathbf{z}_i\mathbf{z}_i^\mathsf{T})] + O(p^{-1/2})$ according to (7) and Assumption 1). This makes the proposed TRFs and its limiting kernel $\mathbf{K}^{ter}$ data-statistics-dependent. Yet, the system of equations (10) does not have a closed-form solution and might have multiple solutions on the real line; we practically solve it using a numerical least squares method, by gradually enlarging the search range (from say $[-1, 1]$) until a solution is found (the time complexity of which is independent of the dimension $n, p$ and it is observed in the experiments in Section 4 to converge in a few iterations). The details of the proposed TRF approach are described in Algorithm 1.

---

**Algorithm 1** Ternary Random Features

---

  **Input:** Data $\mathbf{X}$ and level of sparsity $\epsilon \in [0, 1)$.
  **Output:** Ternary Random Features $\boldsymbol{\Sigma}^{ter}$ and Gram matrix $\mathbf{G}^{ter}$.
  Estimate $\tau$ as $\hat{\tau} = \frac{1}{n}\sum_{i=1}^n \|\mathbf{x}_i\|^2$.
  Solve for thresholds $\hat{s}_-$, $\hat{s}_+$ using (10), which defines $\sigma^{ter}$ via (5).
  Construct a random matrix $\mathbf{W}^{ter} \in \mathbb{R}^{m \times p}$ having i.i.d. entries distributed according to (4).
  Compute $\boldsymbol{\Sigma}^{ter} = \sigma^{ter}(\mathbf{W}^{ter}\mathbf{X})$ and then TRFs Gram matrix $\mathbf{G}^{ter} = \frac{1}{m}(\boldsymbol{\Sigma}^{ter})^\mathsf{T}\boldsymbol{\Sigma}^{ter}$ as in (6).

---

[6]The parameter $\lambda$ characterizes the possibly different $d_0$ between $\mathbf{K}$ and $\mathbf{K}^{ter}$, see details in Appendix A.4.

**Computational and storage complexity**  For $\mathbf{W} \in \mathbb{R}^{m \times p}$ a random matrix with i.i.d. $\mathcal{N}(0,1)$ entries and $\mathbf{W}^{ter} \in \mathbb{R}^{m \times p}$ with i.i.d. entries satisfying (4) with sparsity level $\epsilon \in [0,1)$, let $\mathbf{G} = \frac{1}{m}\sigma(\mathbf{WX})^\mathsf{T}\sigma(\mathbf{WX})$ for some given smooth function $\sigma$ (e.g., sine and cosine in the case of random Fourier features) and $\mathbf{G}^{ter} = \frac{1}{m}\sigma^{ter}(\mathbf{W}^{ter}\mathbf{X})^\mathsf{T}\sigma^{ter}(\mathbf{W}^{ter}\mathbf{X})$ with data matrix $\mathbf{X} \in \mathbb{R}^{p \times n}$. It is then beneficial to use $\mathbf{G}^{ter}$ instead of $\mathbf{G}$ as the computation of $\sigma(\mathbf{WX})$ requires $O(mnp)$ multiplications and $O(mnp)$ additions, while the computation of $\sigma^{ter}(\mathbf{W}^{ter}\mathbf{X})$ requires no multiplication and only $O((1-\epsilon)mnp)$ additions. In terms of storage, it requires a factor of $b = 32$ times more bits to store $\sigma(\mathbf{WX})$ when computing $\mathbf{G}$ compared to storing $\sigma^{ter}(\mathbf{W}^{ter}\mathbf{X})$ when computing $\mathbf{G}^{ter}$ (assuming full precision numbers are stored using $b = 32$ bits).

The computationally and storage efficient TRFs $\sigma^{ter}(\mathbf{W}^{ter}\mathbf{X})$ can then be used instead of the "expensive" random features $\sigma(\mathbf{WX})$ and lead (asymptotically) to the same performance as the latter on downstream tasks, at least for GMM data, according to Theorem 1. In the following section, we provide empirical results showing that (i) this "performance match" between TRFs and random features of the form (1) is *not* limited to Gaussian data and empirically holds when applied on popular real-world datasets such as MNIST (LeCun et al., 1998), some UCI ML datasets, as well as DNN-features of CIFAR10 data in Section A.6 of the appendix; and (ii) due to the competitive performance of TRFs with respect to standard random features approaches, when compared to state-of-the-art random feature compression/quantization techniques (for which a "performance-complexity tradeoff" generally arises, that is, as one compresses more the original random features, the performance decays), TRFs yield significantly better performances for a given storage/computational budget.

## 4 EXPERIMENTS

The experiments in this section and Section A.6 of the appendix are performed on a Ubuntu $18.04$ machine with Intel(R) Core(TM) i9-9900X CPU @ 3.50GHz and $64$ GB of RAM.

### 4.1 TRFS MATCH THE PERFORMANCE OF POPULAR KERNELS WITH LESS BUDGET

We first consider ridge regression with random features Gram matrix $\mathbf{G}$ on MNIST data in Figure 3. We compare RFFs with $\sigma(t) = [\cos(t), \sin(t)]$ and Gaussian $W_{ij} \sim \mathcal{N}(0,1)$ to the proposed TRF method with $\sigma^{ter}(t)$ in (5) and ternary random projection matrix $\mathbf{W}^{ter}$ defined in (4), with different sparsity levels $\epsilon$. The thresholds $s_-, s_+$ of $\sigma^{ter}$ are tuned in such away that the generalized Gaussian moments $d_1$ and $d_2$ match with those of RFFs,[7] as described in Corollary 1 and Algorithm 1. Figure 3 displays the test mean squared errors as a function of the regularization parameter $\gamma$ for our TRF method with different sparsity levels $\epsilon$ compared to the RFF method, as well as to the baseline of Kernel Ridge Regression (KRR) using Gaussian kernel. Note that despite $90\%$ sparsity in the projection matrix $\mathbf{W}$, virtually no performance loss is incurred compared with RFFs. This is further confirmed in the right hand side of Figure 3 which shows the gains in running time[8] when using TRFs. These experiments show that the proposed TRF approach yields similar performance as popular random features such as RFFs, with a significant reduction in terms of computation and storage.

### 4.2 COMPUTATIONAL AND STORAGE GAINS – COMPARISONS TO STATE-OF-THE-ART

In this section, we compare TRFs with state-of-the-art quantized or non-quantized random features methods, for both random features-based logistic and ridge regressions. Specifically, in Figure 1, we compare logistic regression performance of TRFs versus the Low Precision Random Fourier Features (LP-RFFs) proposed by Zhang et al. (2019), on the Cov-Type dataset from UCI ML repo. As in Section 4.1, the TRFs are tuned to match the limiting Gaussian kernel of RFFs. For a single datum $\mathbf{x} \in \mathbb{R}^p$, the associated TRFs $\sigma^{ter}(\mathbf{W}^{ter}\mathbf{x}) \in \mathbb{R}^m$ use $m$ bits for storage while LP-RFFs use $8m$ bits. We follow the same protocol as in (Zhang et al., 2019) and use SGD with mini-batch size 250 in training logistic regressor and $\{1,8\}$ bits precision for the LP-RFF approach. Figure 1 compares the

---

[7]For given $\mathbf{x}_i, \mathbf{x}_j$, we use $[\cos(\mathbf{Wx}_i), \sin(\mathbf{Wx}_j)]$ as the random Fourier features so that the $(i,j)$ entry of the corresponding random features Gram matrix is $\cos(\mathbf{Wx}_i)^\mathsf{T}\cos(\mathbf{Wx}_j) + \sin(\mathbf{Wx}_i)^\mathsf{T}\sin(\mathbf{Wx}_j)$ (Rahimi & Recht, 2008). The generalized Gaussian moments of the RFFs are thus the sum of the $d_1$'s and $d_2$'s corresponding to sin and cos functions. Note that the $d_0$'s of RFFs and TRFs may be different.

[8]The running time is taken as the total clock-time of the whole ridge regression solver including the random features calculation.

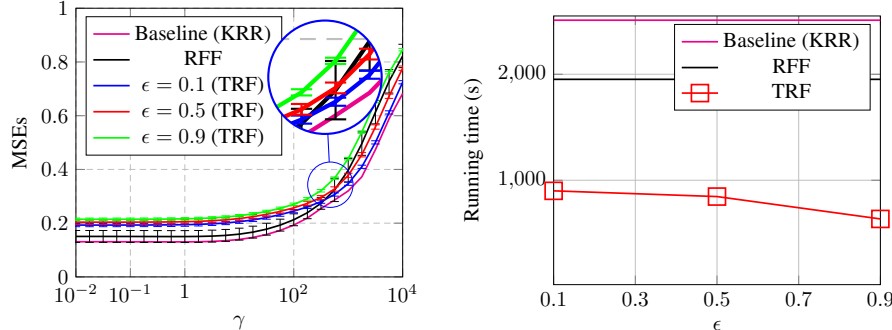

Figure 3: Testing mean squared errors (MSEs with $\pm 1$ std, **LEFT**) and running time (**RIGHT**) of random features ridge regression as a function of regularization parameter $\gamma$, $p = 512, n = 1024, n_{test} = 512, m = 5.10^4$. With **W** distributed according to (4) and $\epsilon = [0.1, \ 0.5, \ 0.9]$, versus RFFs and KRR on a 2-class MNIST dataset – digits $(7, 9)$. We find that $d_0 = 0.44$ for TRFs and $d_0 = 0.39$ for RFFs, making $\lambda$ in Corollary 1 small. Results averaged over 5 independent runs.

logistic regression test accuracy as a function of the total memory budget for LP-RFF (1 bit and 8 bits) and the Nyström approximation of Gaussian kernel matrices (using 32 and 16 bits) (Williams & Seeger, 2001) (see also Table 1 in (Zhang et al., 2019)), versus the proposed TRFs approach. As seen from the shift in the x-axis (memory), by using $8\times$ less memory than LP-RFFs and $32\times$ or $16\times$ less memory than the Nyström method, TRFs achieve a superior generalization performance on the logistic regression task. Similar comparisons are performed in Figure 4 on a random features ridge regression task for the Census dataset (Rahimi & Recht, 2008), confirming that TRFs outperform alternative approaches in terms of test mean square errors, with a significant save in memory.

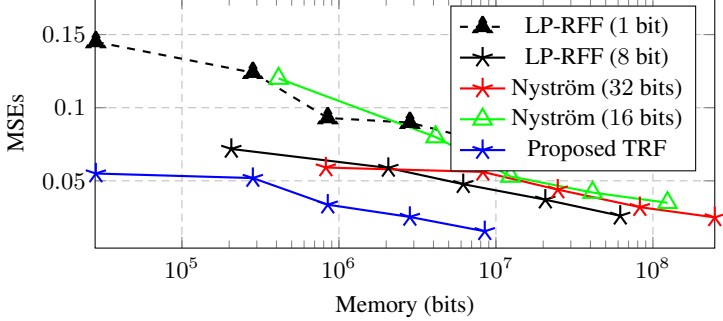

Figure 4: Test mean square errors of ridge regression on quantized random features for different number of features $m \in \{5.10^2, 10^3, 5.10^3, 10^4, 5.10^4\}$, using LP-RFF (Zhang et al., 2019), Nyström approximation (Williams & Seeger, 2001), versus the proposed TRF approach, on the Census dataset, with $n = 16\,000$ training samples, $n_{test} = 2\,000$ test samples, and data dimension $p = 119$.

## 5 CONCLUSION

Our large dimensional spectral analysis of the random features kernel matrix **K** reveals that its spectral properties only depend on the nonlinear activation through the corresponding generalized Gaussian moments and are universal with respect to zero-mean and unit-variance random projection vectors. This allows us to design the new TRF approach which turns both the random weights **W** and the activations $\sigma(\mathbf{WX})$ into ternary integers, thereby allowing to only perform addition operations and to only store 1 bit for the activations. This drastically saves the storage and computation of random features while preserving the performances on downstream tasks with respect to their counterpart expensive kernels. Our article comes along with (Couillet et al., 2021) as first steps in re-designing machine learning algorithms using Random Matrix Theory, in order to be able to perform computations on massive data using desktop computers instead of relying on energy consuming giant servers.

## ACKNOWLEDGEMENT

Z L would like to acknowledge the National Natural Science Foundation of China (NSFC-12141107), the Fundamental Research Funds for the Central Universities of China (2021XXJS110), and CCF-Hikvision Open Fund (20210008) for providing partial support. R C would like to acknowledge the MIAI LargeDATA chair (ANR-19-P3IA-0003) at University Grenobles-Alpes, the HUAWEI LarDist project, as well as the ANR DARLING project for providing partial support of this work.

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

## A  APPENDIX

We provide detailed discussions on our working assumptions in Section A.1, a consistent estimator of the key parameter $\tau$ in Section A.2, the proof of Theorem 1 in Section A.3 and that of Corollary 1 in Section A.4, as well as the generalized Gaussian moments for standard kernels arising from random features and neural net contexts in Section A.5. Additional numerical experiments are placed in Section A.6.

### A.1  NOTES ON THE WORKING ASSUMPTIONS

For completeness, let us restate the working settings of the paper.

Let $\mathbf{x}_1, \cdots, \mathbf{x}_n \in \mathbb{R}^p$ be $n$ independent vectors belonging to one of $K$ distributional classes $\mathcal{C}_1, \cdots, \mathcal{C}_K$. Class $a$ has cardinality $n_a$, and we assume that $\mathbf{x}_i$ follows a Gaussian Mixture Model (GMM), i.e., for $\mathbf{x}_i \in \mathcal{C}_a$,

$$\mathbf{x}_i = \boldsymbol{\mu}_a/\sqrt{p} + \mathbf{z}_i \tag{12}$$

with $\mathbf{z}_i \sim \mathcal{N}(\mathbf{0}_p, \mathbf{C}_a/p)$ for some mean $\boldsymbol{\mu}_a \in \mathbb{R}^p$ and covariance $\mathbf{C}_a \in \mathbb{R}^{p \times p}$ associated to class $\mathcal{C}_a$.

We position ourselves in the non-trivial regime of high-dimensional classification as described by the following growth rate conditions.

**Assumption 4 (High-dimensional asymptotics)** *As $n \to \infty$, we have (i) $p/n \to c \in (0, \infty)$ and $n_a/n \to c_a \in (0, 1)$; (ii) $\|\boldsymbol{\mu}_a\| = O(1)$; (iii) for $\mathbf{C}^\circ = \sum_{a=1}^{K} \frac{n_a}{n} \mathbf{C}_a$ and $\mathbf{C}_a^\circ = \mathbf{C}_a - \mathbf{C}^\circ$, then $\|\mathbf{C}_a\| = O(1)$, $\mathrm{tr}(\mathbf{C}_a^\circ) = O(\sqrt{p})$ and $\mathrm{tr}(\mathbf{C}_a\mathbf{C}_b) = O(p)$ for $a, b \in \{1, \cdots, K\}$. We denote $\tau \triangleq \mathrm{tr}(\mathbf{C}^\circ)/p$ that is assumed to converge in $(0, \infty)$.*

**Beyond Gaussian mixtures**  While the theoretical results in this paper are derived for Gaussian mixture data under the non-trivial setting, we conjecture that they can be extended to a much broader family of data distributions beyond the Gaussian setting, e.g., to the so-called *concentrated random vector* family (Seddik et al., 2020), under similar non triviality assumptions.

**Definition 1 (Concentrated vector)** *Given a normed vector space $(\mathcal{X}, |\cdot|)$, and $q > 0$, a random vector $\mathbf{x} \in \mathcal{X}$ is said to be $q$-exponentially concentrated if for any 1-Lipschitz function $\phi : \mathcal{X} \to \mathbb{R}$, there exists $C > 0$ independent of $dim(\mathcal{X})$ and $\sigma > 0$ such that for all $t \geq 0$,*

$$\mathbb{P}\left(|\phi(\mathbf{x}) - \mathbb{E}[\phi(\mathbf{x})])| > t\right) \leq Ce^{-(t/\sigma)^q}.$$

Multivariate Gaussian distributed $\mathbf{x} \sim \mathcal{N}(\mathbf{0}, \mathbf{I}_p)$ can be checked to belong to the family of concentrated random vectors. The major advantage of using concentrated random vectors for data modeling is that this concentration property is stable under *Lipschitz* transformation, that is, for any 1-Lipschitz map $f : \mathbb{R}^p \to \mathbb{R}^q$, if $\mathbf{x} \in \mathbb{R}^p$ is concentrated, so is $\mathbf{x}' = f(\mathbf{x})$. Among the broad family of concentrated random vectors, it has been particularly shown in (Seddik et al., 2020) that artificial images generated by a Generative Adversarial Network (GAN) (Seddik et al., 2020), which look extremely close to real-world images (as they are designed to), are Lipshitz transformations of Gaussian vectors and can thus be, by definition, concentrated random vectors.

As a result, concentrated random vectors are more appropriate (than GMM for instance) in modeling realistic data, at least for those "close" to data generated by GANs.

The same was shown experimentally for CNN representations of real images (Seddik et al., 2019; 2020) as well as words embeddings in Natural Language Processing (Couillet et al., 2020, November).

As for the random projector matrix, we assume that the matrix $\mathbf{W}$ has i.i.d. entries of zero mean, unit variance and bounded fourth-order moment, with no restriction on their particular distribution as follows.

**Assumption 5 (On random projection matrix)** *The random matrix $\mathbf{W}$ has i.i.d. entries such that $\mathbb{E}[W_{ij}] = 0$, $\mathbb{E}[W_{ij}^2] = 1$ and $\mathbb{E}[W_{ij}^4] < \lambda_W$ for some constant $\lambda_W < \infty$.*

**Remark 2 (On the i.i.d. assumption of entries of $\mathbf{W}$)** *In Assumption 5 we consider the setting where the entries of $\mathbf{W}$ are independently and identically distributed: this is the case for the popular vanilla random Fourier features in (Rahimi & Recht, 2008) and the arc-cosine kernels in a neural network context; but not the case for, e.g., data dependent random features approaches such as leverage score based methods (Li et al., 2021).*

We consider the family of activation functions $\sigma(\cdot)$ satisfying the following assumption.

**Assumption 6 (On activation function)** *The function $\sigma$ is at least twice differentiable (in the sense of distributions when applied to a random variable having a non-degenerate distribution function), with $\max\{\mathbb{E}|\sigma(z)|, \mathbb{E}|\sigma^2(z)|, \mathbb{E}|\sigma'(z)|, \mathbb{E}|\sigma''(z)|\} < \lambda_\sigma$ for some constant $\lambda_\sigma < \infty$ and $z \sim \mathcal{N}(0, 1)$.*

**Remark 3 (Activation functions not differentiable everywhere)** *Some popular activation functions used in machine learning such as ReLu, Sign, Absolute value, etc., are not differentiable everywhere. For those functions, we will use a derivative in the sense of distributions (Friedlander et al., 1998). A distribution $g$ is a continuous linear functional on the set $\mathcal{D}$ of infinitely differentiable functions with bounded support*

$$g : \mathcal{D} \to \mathbb{R}$$

$$\phi \mapsto g(\phi) = \int_{-\infty}^{+\infty} g(x)\phi(x)\, \mathrm{d}x.$$

*The distributional derivative $g'(\phi)$ is defined such that $g'(\phi) = -g(\phi')$. In particular, we will be interested here in the expectation of the derivatives of the activation function with respect to the Gaussian measure i.e., $\int_{-\infty}^{+\infty} \sigma'(x)e^{-x^2/2}\, \mathrm{d}x$. Following the previous definition, we have in this particular case*

$$\int_{-\infty}^{+\infty} \sigma'(x)e^{-x^2/2}\, \mathrm{d}x = \int_{-\infty}^{+\infty} x\sigma(x)e^{-x^2/2}\, \mathrm{d}x$$

*which can be evaluated by some integration by parts. This is also refereed to the as "weak derivative" in the functional analysis literature (Stein & Shakarchi, 2012).*

## A.2 AUXILIARY RESULTS AND PROOFS

**Lemma 1 (Consistent estimation of $\tau$)** *Let Assumption 4 hold and define $\tau \triangleq \mathrm{tr}\mathbf{C}^\circ/p$. Then as $n \to \infty$, with probability 1,*

$$\frac{1}{n}\sum_{i=1}^{n}\|\mathbf{x}_i\|^2 - \tau \to 0.$$

**Proof 1 (Proof of Lemma 1)** *From equation 7, we have that*

$$\frac{1}{n}\sum_{i=1}^{n}\|\mathbf{x}_i\|^2 = \frac{1}{n}\sum_{a=1}^{K}\sum_{i=1}^{n}\frac{1}{p}\|\boldsymbol{\mu}_a\|^2 - \frac{2}{\sqrt{p}}\boldsymbol{\mu}_a^\mathsf{T}\mathbf{z}_i + \frac{1}{n}\sum_{i=1}^{n}\|\mathbf{z}_i\|^2. \tag{13}$$

*From Assumption 4, we have that $\frac{1}{n}\sum_{a=1}^{K}\sum_{i=1}^{n}\frac{1}{p}\|\boldsymbol{\mu}_a\|^2 = O(p^{-1})$. The second term of equation 13 $\frac{2}{\sqrt{p}}\boldsymbol{\mu}_a^\mathsf{T}\mathbf{z}_i$ is a weighted sum of independent zero mean random variables; it thus vanishes with probability 1 as $n, p \to \infty$ by a mere application of Chebyshev's inequality and the Borell Cantelli lemma. Finally, using the strong law of large numbers on the last term of equation 13, we have almost surely,*

$$\frac{1}{n}\sum_{i=1}^{n}\|\mathbf{z}_i\|^2 = \sum_{a=1}^{K}\frac{n_a}{n}\mathrm{tr}\mathbf{C}_a + o(1)$$

$$= \sum_{a=1}^{K}\frac{n_a}{n}\frac{1}{p}\mathrm{tr}\mathbf{C}^\circ + o(1)$$

*where in the last line we use $\mathrm{tr}\mathbf{C}_a^\circ = O(\sqrt{p})$ from Assumption 4, and thus $\frac{1}{n}\sum_{i=1}^{n}\|\mathbf{z}_i\|^2 - \tau \to 0$ almost surely. This concludes the proof.*

## A.3 PROOF OF THEOREM 1

In the sequel, we will make use of Bachmann–Landau notations but specific for random variables in the asymptotic regime where $n \to \infty$. For $x \equiv x_n$ a random variable and $u_n \geq 0$, we write $x_n = O(u_n)$ if for any $\eta$ and $D > 0$, $n^D\mathbb{P}(x \geq n^\eta u_n) \to 0$ as $n \to \infty$. For a vector $\mathbf{v}$ or a diagonal matrix with random entries, $\mathbf{v} = O(u_n)$ means that the maximal entry of $\mathbf{v}$ in absolute value is $O(u_n)$ in the sense defined above. When $\mathbf{M}$ is a square matrix, $\mathbf{M} = O(u_n)$ means that the operator norm of $\mathbf{M}$ is $O(u_n)$. For $x$ a random variable, $x = o(u_n)$ means that there exists $\kappa > 0$ such that $x = O(n^{-\kappa}u_n)$. And the same definition for the $o(\cdot)$ notation applies for vectors and matrices having random entries.

Let us define $\mathbf{X} = [\mathbf{x}_1, \cdots, \mathbf{x}_n] \in \mathbb{R}^{p \times n}$ and define

$$\boldsymbol{\Sigma} = \sigma(\mathbf{W}\mathbf{X})$$

where $\mathbf{W} = [\mathbf{w}_1, \cdots, \mathbf{w}_m]^\mathsf{T} \in \mathbb{R}^{m \times p}$ with $\mathbf{W}$ satisfying Assumption 5, and $\sigma$ a function satisfying Assumption 6. We consider the gram matrix

$$\mathbf{G} = \frac{1}{m}\boldsymbol{\Sigma}^\mathsf{T}\boldsymbol{\Sigma}$$

whose $(i, j)$ entry is given by

$$G_{ij} = \frac{1}{m}\sum_{k=1}^{m}\sigma(\mathbf{w}_k^\mathsf{T}\mathbf{x}_i)\sigma(\mathbf{w}_k^\mathsf{T}\mathbf{x}_j).$$

We are interested in computing

$$\kappa(\mathbf{x}_i, \mathbf{x}_j) = \mathbb{E}_\mathbf{w}[\sigma(\mathbf{w}^\mathsf{T}\mathbf{x}_i)\sigma(\mathbf{w}^\mathsf{T}\mathbf{x}_j)] \tag{14}$$

under Assumptions 4-6.

Under Assumption 4, we have

$$\mathbf{x}_i^\mathsf{T} \mathbf{x}_j = \underbrace{\mathbf{z}_i^\mathsf{T} \mathbf{z}_j}_{O(p^{-\frac{1}{2}})} + \underbrace{\boldsymbol{\mu}_a^\mathsf{T} \boldsymbol{\mu}_b / p + \boldsymbol{\mu}_a^\mathsf{T} \mathbf{w}_j / \sqrt{p} + \boldsymbol{\mu}_b^\mathsf{T} \mathbf{w}_i / \sqrt{p}}_{O(p^{-1})} \tag{15}$$

and

$$\|\mathbf{x}_i\|^2 = \underbrace{\tau}_{O(1)} + \underbrace{\mathrm{tr}(\mathbf{C}_a^\circ)/p + \Phi_i}_{O(p^{-\frac{1}{2}})} + \underbrace{\|\boldsymbol{\mu}_a^2\|/p + 2\boldsymbol{\mu}_a^\mathsf{T} \mathbf{z}_i / \sqrt{p}}_{O(p^{-1})} \tag{16}$$

where $\phi_i \triangleq (\|\mathbf{z}_i\|^2 - \mathbb{E}[\|\mathbf{z}_i\|^2])$.

It can be checked that, for random vector $\mathbf{w} \in \mathbb{R}^p$ having i.i.d. entries with $\mathbb{E}[w_i] = 0$ and $\mathbb{E}[w_i^2] = 1$, we have, conditioned on $\mathbf{x}_i$, that $\mathbb{E}_\mathbf{w}[(\mathbf{w}^\mathsf{T} \mathbf{x}_i)^2] = \|\mathbf{x}_i\|^2$ and

$$\mathbb{E}_\mathbf{w}[(\mathbf{w}^\mathsf{T} \mathbf{x}_i)^4] = (m_4 - 3)\|\mathbf{x}_i\|^2 + 2\|\mathbf{x}_i\|^4. \tag{17}$$

with $m_4 = \mathbb{E}[w_j^4] < \lambda_W$ according to Assumption 5. From the trace lemma, (Bai & Silverstein, 2010, Lemma B.26), one has that

$$\mathbf{x}_i^\mathsf{T} \mathbf{x}_i - \frac{1}{p}\mathrm{tr}\mathbf{C}_a \to 0 \tag{18}$$

almost surely as $p \to \infty$, for $\mathbf{x}_i \in \mathcal{C}_a$. Thus, under Assumption 4 we have in particular $\mathbf{x}_i^\mathsf{T} \mathbf{x}_i - \mathrm{tr}\mathbf{C}^\circ/p \to 0$ holds almost surely, regardless of the class of $\mathbf{x}_i$, see the proof of Lemma 1. It then follows from Lyapunov CLT (see, e.g., (Billingsley, 2012, Theorem 27.3)) that, under Assumption 4 and 5, $(\mathbf{w}^\mathsf{T} \mathbf{x}_i, \mathbf{w}^\mathsf{T} \mathbf{x}_j)$ is asymptotically bivariate Gaussian. We can thus perform, in the large $p$ limit, a Gram-Schmidt orthogonalization procedure for some standard Gaussian variables $\xi_a$, $\xi_b \sim \mathcal{N}(0, 1)$. By construction, $\xi_a$, $\xi_b$ are *uncorrelated* and thus *independent* by Gaussianity. Let us denote the shortcuts $\zeta_a = \mathbf{w}^\mathsf{T} \mathbf{x}_i$ and $\zeta_b = \mathbf{w}^\mathsf{T} \mathbf{x}_j$. We thus have

$$\zeta_a \equiv \mathbf{w}^\mathsf{T} \mathbf{x}_i = u_a \xi_a + o(1)$$
$$\zeta_b \equiv \mathbf{w}^\mathsf{T} \mathbf{x}_j = v_b \xi_a + u_b \xi_b + o(1)$$

with

$$u_a = \|\mathbf{x}_i\|$$
$$v_b = \frac{\mathbf{x}_i^\mathsf{T} \mathbf{x}_j}{\|\mathbf{x}_i\|}$$
$$u_b = \sqrt{\|\mathbf{x}_j\|^2 - \frac{(\mathbf{x}_i^\mathsf{T} \mathbf{x}_j)^2}{\|\mathbf{x}_i\|^2}}.$$

Since we have that $\|\mathbf{x}_i\| = \sqrt{\tau} + o(1)$ and $\|\mathbf{x}_j\| = \sqrt{\tau} + o(1)$ (from equation 16), we can perform a Taylor expansion of $\sigma(\zeta_a)$ (respectively of $\sigma(\zeta_b)$) around $\sqrt{\tau}\xi_a$ (resp. $\sqrt{\tau}\xi_b$) giving

$$\sigma(\zeta_a) = \sigma(\sqrt{\tau}\xi_a) + \sigma'(\sqrt{\tau}\xi_a)(\zeta_a - \sqrt{\tau}\xi_a) + \frac{1}{2}\sigma''(\xi_a)(\zeta_a - \sqrt{\tau}\xi_a)^2 + o((\zeta_a - \sqrt{\tau}\xi_a)^2)$$

$$\sigma(\zeta_b) = \sigma(\sqrt{\tau}\xi_b) + \sigma'(\sqrt{\tau}\xi_b)(\zeta_b - \sqrt{\tau}\xi_b) + \frac{1}{2}\sigma''(\sqrt{\tau}\xi_b)(\zeta_b - \sqrt{\tau}\xi_b)^2 + o((\zeta_b - \sqrt{\tau}\xi_b)^2).$$

We then have

$$\kappa(\mathbf{x}_i, \mathbf{x}_j) = \mathbb{E}[\sigma(\zeta_a)\sigma(\zeta_b)]$$
$$= \mathbb{E}[\sigma(\sqrt{\tau}\xi_a)\sigma(\sqrt{\tau}\xi_b)] + \mathbb{E}[\sigma(\sqrt{\tau}\xi_a)\sigma'(\sqrt{\tau}\xi_b)(\zeta_b - \sqrt{\tau}\xi_b)]$$
$$+ \frac{1}{2}\mathbb{E}[\sigma(\sqrt{\tau}\xi_a)\sigma''(\sqrt{\tau}\xi_b)(\zeta_b - \sqrt{\tau}\xi_b)^2]$$
$$+ \mathbb{E}[\sigma'(\sqrt{\tau}\xi_a)(\zeta_a - \sqrt{\tau}\xi_a)\sigma(\sqrt{\tau}\xi_b)] + \mathbb{E}[\sigma'(\sqrt{\tau}\xi_a)(\zeta_a - \sqrt{\tau}\xi_a)\sigma'(\sqrt{\tau}\xi_b)(\zeta_b - \sqrt{\tau}\xi_b)]$$
$$+ \frac{1}{2}\mathbb{E}[\sigma'(\xi_a)(\zeta_a - \sqrt{\tau}\xi_a)\sigma''(\xi_b)(\zeta_b - \sqrt{\tau}\xi_b)^2] + \frac{1}{2}\mathbb{E}[\sigma''(\xi_a)(\zeta_a - \sqrt{\tau}\xi_a)^2\sigma(\sqrt{\tau}\xi_b)]$$
$$+ \frac{1}{2}\mathbb{E}[\sigma''(\sqrt{\tau}\xi_a)(\zeta_a - \sqrt{\tau}\xi_a)^2\sigma'(\sqrt{\tau}\xi_b)(\zeta_b - \sqrt{\tau}\xi_b)]$$
$$+ \frac{1}{4}\mathbb{E}[\sigma''(\sqrt{\tau}\xi_a)(\zeta_a - \sqrt{\tau}\xi_a)^2\sigma''(\sqrt{\tau}\xi_b)(\zeta_b - \sqrt{\tau}\xi_b)^2] + o((\zeta_a - \sqrt{\tau}\xi_a)^2).$$

where the expectation is with respect to $\mathbf{w}$ as in the definition in (14).

Using the independence between $\xi_a, \xi_b$ along with the following

$$\zeta_a - \sqrt{\tau}\xi_a = (\frac{u_a}{\sqrt{\tau}} - 1)\sqrt{\tau}\xi_a \tag{19}$$

$$\zeta_b - \sqrt{\tau}\xi_b = (\frac{u_b}{\sqrt{\tau}} - 1)\sqrt{\tau}\xi_b + \frac{v_b}{\sqrt{\tau}}\sqrt{\tau}\xi_a \tag{20}$$

we have for $\xi \sim \mathcal{N}(0,1)$,

$$\mathbb{E}[\sigma(\sqrt{\tau}\xi_a)\sigma(\sqrt{\tau}\xi_b)] = \left(\mathbb{E}[\sigma(\sqrt{\tau}\xi)]\right)^2 \tag{21}$$

$$\mathbb{E}[\sigma(\sqrt{\tau}\xi_a)\sigma'(\sqrt{\tau}\xi_b)(\zeta_b - \sqrt{\tau}\xi_b)] = \left(\mathbb{E}[\sigma(\sqrt{\tau}\xi)]\mathbb{E}[\sqrt{\tau}\xi\sigma'(\sqrt{\tau}\xi)]\right)\left(\frac{u_b}{\sqrt{\tau}} - 1\right)$$
$$+ \left(\mathbb{E}[\sigma'(\sqrt{\tau}\xi)]\mathbb{E}[\sqrt{\tau}\xi\sigma(\sqrt{\tau}\xi)]\right)\frac{v_b}{\sqrt{\tau}} \tag{22}$$

Similarly writing down the products $(\zeta_a - \sqrt{\tau}\xi_a)^2, (\zeta_b - \sqrt{\tau}\xi_b)^2, (\zeta_a - \sqrt{\tau}\xi_a)(\zeta_b - \sqrt{\tau}\xi_b), (\zeta_a - \sqrt{\tau}\xi_a)(\zeta_b - \sqrt{\tau}\xi_b)^2, (\zeta_a - \sqrt{\tau}\xi_a)^2(\zeta_b - \sqrt{\tau}\xi_b), (\zeta_a - \sqrt{\tau}\xi_a)^2(\zeta_b - \sqrt{\tau}\xi_b)^2$ as in Equations 21- 22, and using Equations 19- 20, we obtain a complete expression of all the terms in $\kappa(\mathbf{x}_i, \mathbf{x}_j)$ as follows

$$\kappa(\mathbf{x}_i, \mathbf{x}_j) = \left[\left(\mathbb{E}[\sigma(\sqrt{\tau}\xi)]\right)^2 + \left(\mathbb{E}[\sigma(\sqrt{\tau}\xi)]\mathbb{E}[\sqrt{\tau}\xi\sigma'(\sqrt{\tau}\xi)]\right)\left(\frac{u_b}{\sqrt{\tau}} - 1\right) + \left(\mathbb{E}[\sigma'(\sqrt{\tau}\xi)]\mathbb{E}[\sqrt{\tau}\xi\sigma(\sqrt{\tau}\xi)]\right)\frac{v_b}{\sqrt{\tau}}\right.$$

$$+ \left(\mathbb{E}[\sqrt{\tau}\xi\sigma'(\sqrt{\tau}\xi)]^2\right)\left((\frac{u_a}{\sqrt{\tau}} - 1)(\frac{u_b}{\sqrt{\tau}} - 1)\right) + \frac{\left(\mathbb{E}[\sigma(\sqrt{\tau}\xi)]\mathbb{E}[\tau\xi^2\sigma''(\sqrt{\tau}\xi)]\right)}{2}\left((u_b - 1)^2\right)$$

$$+ \frac{\left(\mathbb{E}[\sqrt{\tau}\xi\sigma'(\sqrt{\tau}\xi)]\mathbb{E}[\tau\xi^2\sigma''(\sqrt{\tau}\xi)]\right)}{2}\left(\frac{u_a}{\sqrt{\tau}} - 1\right)\left(\frac{u_b}{\sqrt{\tau}} - 1\right)^2$$

$$+ \frac{\left(\mathbb{E}[\sqrt{\tau}\xi\sigma'(\sqrt{\tau}\xi)]\mathbb{E}[\tau\xi^2\sigma''(\sqrt{\tau}\xi)]\right)}{2}\left(\frac{u_a}{\sqrt{\tau}} - 1\right)^2\left(\frac{u_b}{\sqrt{\tau}} - 1\right)$$

$$+ \frac{\left(\mathbb{E}[\tau\xi^2\sigma''(\sqrt{\tau}\xi)]^2\right)}{4}\left(\frac{u_a}{\sqrt{\tau}} - 1\right)^2\left(\frac{u_b}{\sqrt{\tau}} - 1\right)^2 + \left(\mathbb{E}[\sqrt{\tau}\xi\sigma(\sqrt{\tau}\xi)]\mathbb{E}[\sqrt{\tau}\xi\sigma''(\sqrt{\tau}\xi)]\right)\frac{v_b}{\sqrt{\tau}}\left(\frac{u_b}{\sqrt{\tau}} - 1\right)$$

$$+ \frac{\left(\mathbb{E}[\sigma''(\sqrt{\tau}\xi)]\mathbb{E}[\tau\xi^2\sigma''(\sqrt{\tau}\xi)]\right)}{2}\left(\frac{v_b}{\sqrt{\tau}}\right)^2 + \left(\mathbb{E}[\sigma(\sqrt{\tau}\xi)]\mathbb{E}[\sqrt{\tau}\xi\sigma'(\sqrt{\tau}\xi)]\right)\left(\frac{u_a}{\sqrt{\tau}} - 1\right)$$

$$+ \left(\mathbb{E}[\sigma'(\sqrt{\tau}\xi)]\mathbb{E}[\xi^2\sigma'(\sqrt{\tau}\xi)]\right)\frac{v_b}{\sqrt{\tau}}\left(\frac{u_a}{\sqrt{\tau}} - 1\right) + \frac{\left(\mathbb{E}[\sigma''(\sqrt{\tau}\xi)]\mathbb{E}[\tau\sqrt{\tau}\xi^3\sigma'(\sqrt{\tau}\xi)]\right)}{2}\left(\frac{v_b}{\sqrt{\tau}}\right)^2\left(\frac{u_a}{\sqrt{\tau}} - 1\right)$$

$$+ \frac{\left(\mathbb{E}[\sigma'(\sqrt{\tau}\xi)]\mathbb{E}[\tau\xi^2\sigma'(\sqrt{\tau}\xi)]\right)}{2}\frac{v_b}{\sqrt{\tau}}\left(\frac{u_a}{\sqrt{\tau}} - 1\right)\left(\frac{u_b}{\sqrt{\tau}} - 1\right) + \frac{\left(\mathbb{E}[\sigma(\sqrt{\tau}\xi)]\mathbb{E}[\tau\xi^2\sigma''(\sqrt{\tau}\xi)]\right)}{2}\left(\frac{u_a}{\sqrt{\tau}} - 1\right)^2$$

$$+ \frac{\left(\mathbb{E}[\sigma'(\sqrt{\tau}\xi)]\mathbb{E}[\tau\sqrt{\tau}\xi^3\sigma''(\sqrt{\tau}\xi)]\right)}{2}\frac{v_b}{\sqrt{\tau}}\left(\frac{u_a}{\sqrt{\tau}} - 1\right)^2$$

$$+ \frac{\left(\mathbb{E}[\sigma''(\sqrt{\tau}\xi)]\mathbb{E}[\tau^2\xi^4\sigma''(\sqrt{\tau}\xi)]\right)}{4}\left(\frac{v_b}{\sqrt{\tau}}\right)^2\left(\frac{u_a}{\sqrt{\tau}} - 1\right)^2$$

$$\left. + \frac{\left(\mathbb{E}[\sqrt{\tau}\xi\sigma''(\sqrt{\tau}\xi)]\mathbb{E}[\tau\sqrt{\tau}\xi^3\sigma''(\sqrt{\tau}\xi)]\right)}{2}\frac{v_b}{\sqrt{\tau}}\left(\frac{u_b}{\sqrt{\tau}} - 1\right)\left(\frac{u_a}{\sqrt{\tau}} - 1\right)^2\right] + O(p^{-\frac{3}{2}}).$$
$$\tag{23}$$

Since $|\mathbf{x}_i^\mathsf{T}\mathbf{x}_j| \leq \epsilon$ for a sufficiently small $\epsilon$, (following from equation 15), using a Taylor approximation of $\sqrt{\|\mathbf{x}_j\|^2 - \frac{(\mathbf{x}_i^\mathsf{T}\mathbf{x}_j)^2}{\|\mathbf{x}_i\|^2}}$ we obtain

$$\frac{u_b}{\sqrt{\tau}} - 1 = \frac{1}{\sqrt{\tau}}\sqrt{\|\mathbf{x}_j\|^2 - \frac{(\mathbf{x}_i^\mathsf{T}\mathbf{x}_j)^2}{\|\mathbf{x}_i\|^2}} - 1 = \left(\frac{\|x_j\|}{\sqrt{\tau}} - 1\right) - \frac{1}{2\sqrt{\tau}\|\mathbf{x}_j\|}\frac{(\mathbf{x}_i^\mathsf{T}\mathbf{x}_j)^2}{\|\mathbf{x}_i\|^2} + o\left(\frac{(\mathbf{x}_i^\mathsf{T}\mathbf{x}_j)^2}{\|\mathbf{x}_i\|^2}\right)$$

$$\left(\frac{u_b}{\sqrt{\tau}} - 1\right)^2 = \left(\frac{1}{\sqrt{\tau}}\sqrt{\|\mathbf{x}_j\|^2 - \frac{(\mathbf{x}_i^\mathsf{T}\mathbf{x}_j)^2}{\|\mathbf{x}_i\|^2\|\mathbf{x}_j\|^2}} - 1\right)^2 = \left(\frac{\|\mathbf{x}_j\|}{\sqrt{\tau}} - 1\right)^2 - \frac{\frac{\|\mathbf{x}_j\|}{\sqrt{\tau}} - 1}{\|\mathbf{x}_j\|}\frac{(\mathbf{x}_i^\mathsf{T}\mathbf{x}_j)^2}{\|\mathbf{x}_i\|^2} + o\left(\frac{(\mathbf{x}_i^\mathsf{T}\mathbf{x}_j)^2}{\|\mathbf{x}_i\|^2}\right)$$

$$\frac{u_a}{\sqrt{\tau}} - 1 = \frac{\|\mathbf{x}_i\|}{\sqrt{\tau}} - 1.$$

We thus have

$$\frac{v_b}{\sqrt{\tau}} = \frac{\mathbf{x}_i^\mathsf{T}\mathbf{x}_j}{\sqrt{\tau}\|\mathbf{x}_i\|} = \frac{1}{\tau}\left(\underbrace{\mathbf{z}_i^\mathsf{T}\mathbf{z}_j}_{O(p^{-\frac{1}{2}})} + \underbrace{\left(\frac{\boldsymbol{\mu}_a^\mathsf{T}\boldsymbol{\mu}_b}{p} + \frac{\boldsymbol{\mu}_a^\mathsf{T}\mathbf{w}_j}{\sqrt{p}} + \frac{\boldsymbol{\mu}_b^\mathsf{T}\mathbf{w}_i}{\sqrt{p}}\right)}_{O(p^{-1})}\right) + O(p^{-\frac{3}{2}})$$

$$\left(\frac{u_a}{\sqrt{\tau}} - 1\right) = \frac{\|\mathbf{x}_i\|}{\sqrt{\tau}} - 1 = \underbrace{\frac{1}{2\tau}\left(\frac{\mathrm{tr}(\mathbf{C}_a^\circ)}{p} + \Phi_i\right)}_{O(p^{-\frac{1}{2}})} + \underbrace{\frac{1}{2\tau}\left(\frac{\|\boldsymbol{\mu}_a\|^2}{p} + 2\boldsymbol{\mu}_a^\mathsf{T}\frac{\mathbf{z}_i}{\sqrt{p}}\right)}_{O(p^{-1})} + O(p^{-\frac{3}{2}})$$

$$\left(\frac{u_a}{\sqrt{\tau}} - 1\right)\left(\frac{u_b}{\sqrt{\tau}} - 1\right) = \left(\frac{\|\mathbf{x}_i\|}{\sqrt{\tau}} - 1\right)\left(\frac{\|\mathbf{x}_j\|}{\sqrt{\tau}} - 1\right) = \underbrace{\frac{\left(\frac{\mathrm{tr}(\mathbf{C}_a^\circ)}{p} + \Phi_i\right)\left(\frac{\mathrm{tr}(\mathbf{C}_b^\circ)}{p} + \Phi_j\right)}{4\tau^2}}_{O(p^{-\frac{1}{2}})} + O(p^{-\frac{3}{2}})$$

$$\left(\frac{v_b}{\sqrt{\tau}}\right)^2 = \frac{1}{\tau^2}(\mathbf{z}_i^\mathsf{T}\mathbf{z}_j)^2 + O(p^{-\frac{3}{2}}). \tag{24}$$

All other terms in equation 23 are of order at most $O(p^{-\frac{3}{2}})$ and thus vanish asymptotically. We thus get

$$\kappa(\mathbf{x}_i, \mathbf{x}_j) = \left[\left(\mathbb{E}[\sigma(\sqrt{\tau}\xi)]\right)^2 + \left(\mathbb{E}[\sigma(\sqrt{\tau}\xi)]\mathbb{E}[\sqrt{\tau}\xi\sigma'(\sqrt{\tau}\xi)]\right)\left(\frac{u_b}{\sqrt{\tau}} - 1\right) + \left(\mathbb{E}[\sigma'(\sqrt{\tau}\xi)]\mathbb{E}[\sqrt{\tau}\xi\sigma(\sqrt{\tau}\xi)]\right)\frac{v_b}{\sqrt{\tau}} + \right.$$

$$+ \left(\mathbb{E}[\sqrt{\tau}\xi\sigma'(\sqrt{\tau}\xi)]^2\right)\left((\frac{u_a}{\sqrt{\tau}} - 1)(\frac{u_b}{\sqrt{\tau}} - 1)\right) + \frac{\left(\mathbb{E}[\sigma''(\sqrt{\tau}\xi)]\mathbb{E}[\tau\xi^2\sigma(\sqrt{\tau}\xi)]\right)}{2}\left(\frac{v_b}{\sqrt{\tau}}\right)^2$$

$$\left. + \left(\mathbb{E}[\sigma(\sqrt{\tau}\xi)]\mathbb{E}[\sqrt{\tau}\xi\sigma'(\sqrt{\tau}\xi)]\right)\left(\frac{u_a}{\sqrt{\tau}} - 1\right) + O(p^{-\frac{3}{2}})\right]. \tag{25}$$

Plugging the terms in equation 24 into equation 25 and rearranging in matrix form, we obtain for $\mathbf{K} = \mathbf{P}\{\kappa(\mathbf{x}_i, \mathbf{x}_j)\}_{i,j=1}^n\mathbf{P}$ and $\tilde{\mathbf{K}}$ defined in Theorem 1, that $\|\mathbf{K} - \tilde{\mathbf{K}}\| \to 0$, as expected, where we used the fact that $\|\mathbf{A}\| \leq n\|\mathbf{A}\|_\infty$ for $\mathbf{A} \in \mathbb{R}^{n\times n}$ and $\|\mathbf{A}\|_\infty = \max_{i,j=1}^n |\mathbf{A}_{ij}|$. This concludes the proof of Theorem 1.

### A.4 PROOF OF COROLLARY 1

Define the Ternary Random Features matrix $\boldsymbol{\Sigma}^{ter} = \sigma^{ter}(\mathbf{W}^{ter}\mathbf{X})$ with $\mathbf{W}^{ter}$ defined in (4) having sparsity level $\epsilon$, the associated Gram matrix $\mathbf{G}^{ter} = \frac{1}{m}(\boldsymbol{\Sigma}^{ter})^\mathsf{T}\boldsymbol{\Sigma}^{ter}$ as in (6), and the limiting kernel

$$\mathbf{K}^{ter} \triangleq \mathbf{P}\{\kappa^{ter}(\mathbf{x}_i, \mathbf{x}_j)\}_{i,j=1}^n\mathbf{P} \tag{26}$$

for $\kappa^{ter}$ defined in (3). After calculation similar to (Liao & Couillet, 2019, Section 4) and (Liao et al., 2021), we obtain

$$\mathbb{E}[(\sigma^{ter})'(\sqrt{\tau}z)]^2 = \left(\frac{e^{-\frac{s_+^2}{\tau}} + e^{-\frac{s_-^2}{\tau}}}{\pi}\right)^2$$

$$\mathbb{E}[(\sigma^{ter})''(\sqrt{\tau}z)]^2 = \left(\frac{s_+ e^{-\frac{s_+^2}{\tau}} + s_- e^{-\frac{s_-^2}{\tau}}}{\tau\sqrt{2\pi\tau}}\right)^2. \tag{27}$$

Thus, according to Theorem 1, for random features kernel matrix $\mathbf{K}$ of the form (2) with $\mathbf{W}$ and nonlinear $\sigma$ satisfying Assumption 1-3, with associated generalized Gaussian moments $d_0, d_1, d_2$ defined in Theorem 1, by choosing $s_-$, and $s_+$ such that $d_1$ and $d_2$ are respectively equal to the first and second term of equation 27, we have that

$$\|\mathbf{K} - \mathbf{K}^{ter} - \lambda\mathbf{P}\| \to 0,$$

almost surely with

$$\lambda = d_0 - \left(\mathbb{E}[(\sigma^{ter})^2(\sqrt{\tau}z)] - \mathbb{E}[(\sigma^{ter})(\sqrt{\tau}z)]^2 - \tau\mathbb{E}[(\sigma^{ter})'(\sqrt{\tau}z)]^2\right)$$

where

$$\mathbb{E}[(\sigma^{ter})^2(\sqrt{\tau}z)] - \mathbb{E}[(\sigma^{ter})(\sqrt{\tau}z)]^2 - \tau\mathbb{E}[(\sigma^{ter})'(\sqrt{\tau}z)]^2$$

$$= \left(1 - \mathrm{erf}\left(\frac{s_+}{\sqrt{\tau}}\right)\right) - \frac{1}{2}\left(\frac{e^{-\frac{s_+^2}{\tau}} + e^{-\frac{s_-^2}{\tau}}}{\pi}\right)^2.$$

A.5  GAUSSIAN MOMENTS OF POPULAR ACTIVATION FUNCTIONS

We provide in Table 1 the calculation of the generalized Gaussian moments $d_0, d_1, d_2$ for popular activation functions used in random features and neural network contexts. Note that our Table 1 matches the results in (Liao & Couillet, 2018b, Table 2).

Table 1: Values of $d_0, d_1, d_2$ for different activation functions.

| $\sigma(t)$ | $d_0$ | $d_1$ | $d_2$ |
|---|---|---|---|
| $\|t\|$ | $\tau\left(1 - \frac{2}{\pi}\right)$ | $0$ | $\frac{1}{2\pi\tau}$ |
| $\max(0, t)$ | $\frac{\tau}{2}\left(\frac{1}{2} - \frac{1}{\pi}\right)$ | $\frac{1}{4}$ | $\frac{1}{8\pi\tau}$ |
| $\begin{array}{ll} -1, & t < s_- \\ +1, & t > s_+ \\ 0, & otherwise \end{array}$ | $\left(1 - \operatorname{erf}\left(\frac{s_+}{\sqrt{\tau}}\right)\right)$ $-\frac{1}{2}\left(\frac{e^{-\frac{s_+^2}{\tau}} + e^{-\frac{s_-^2}{\tau}}}{\pi}\right)^2$ | $\left(\frac{e^{-\frac{s_+^2}{\tau}} + e^{-\frac{s_-^2}{\tau}}}{\pi}\right)^2$ | $\left(\frac{s_+ e^{-\frac{s_+^2}{\tau}} + s_- e^{-\frac{s_-^2}{\tau}}}{\tau\sqrt{2\pi\tau}}\right)^2$ |
| $a_+\max(0, t) + a_-\max(0, -t)$ | $\tau(a_+ + a_-)^2\left(\frac{\pi - 2}{4\pi}\right)$ | $\frac{(a_+ - a_-)^2}{4}$ | $\frac{(a_+ + a_-)^2}{8\pi\tau}$ |
| $a_2 t^2 + a_1 t + a_0$ | $2\tau^2 a_2^2$ | $a_1^2$ | $a_2^2$ |
| $\exp(t)$ | $\frac{1}{\sqrt{2\tau+1}} - \frac{1}{\tau+1}$ | $0$ | $\frac{1}{4(\tau+1)^3}$ |
| $\cos(t)$ | $\frac{1 + e^{-2\tau}}{2} - e^{-\tau}$ | $0$ | $\frac{e^{-\tau}}{4}$ |
| $\sin(t)$ | $\frac{1 - e^{-2\tau}}{2} - \tau e^{-\tau}$ | $e^{-\tau}$ | $0$ |
| $t$ | $0$ | $1$ | $0$ |
| $\operatorname{sign}(t)$ | $1 - \frac{2}{\pi}$ | $\frac{2}{\pi\tau}$ | $0$ |
| $1_{t>0}$ | $\frac{1}{4} - \frac{1}{2\pi}$ | $\frac{1}{2\pi\tau}$ | $0$ |

A.6  ADDITIONAL EXPERIMENTS

In this section, we complement Section 4 by providing additional experiments on ridge regression and support vector machine, to support the robustness our TRF method compared to state-of-the-art approaches.

**Remark 4 (On the empirical and expected random features kernels)** *It is worth noting that our main results in Theorem 1 and Corollary 1 characterize the behavior of the* expected/*limiting random features kernel* $\mathbf{K}$ *instead of the* empirical *Gram kernel matrix* $\mathbf{G}$ *defined in (6) obtained by averaging over $m$ random features. The operator norm difference $\|\mathbf{K} - \mathbf{G}\|$ is then bound to vanish as $m, p, n \to \infty$ for a sufficiently large $m$ (for example with $m/\max(n, p) \to \infty$).*

**Random features based Ridge regression**  To complement the experiments in Section 4.1, we provide in Figures 5-9, the test mean square error of random features kernel ridge regression with increasing number of random features $m \in \{512, 4096, 10^4\}$ for GMM data in Figure 5-7, and $m \in \{512, 10^4\}$ for MNIST data in Figure 8 and 9, as a function of the regularization parameter $\gamma$ and for different choices of sparsity levels $\epsilon \in \{0.1, 0.3, 0.5, 0.7, 0.9\}$. It is interesting to note that, for both datasets, the performance gap between RFFs and TRFs significantly decreases as $m$ the number of random features grows large: this is in agreement with our Theorem 1 in which guarantee is provided only for the *expected* kernel matrix (that corresponds to $m \to \infty$), not the *empirical* Gram matrix as the sample mean over $m$ random features.

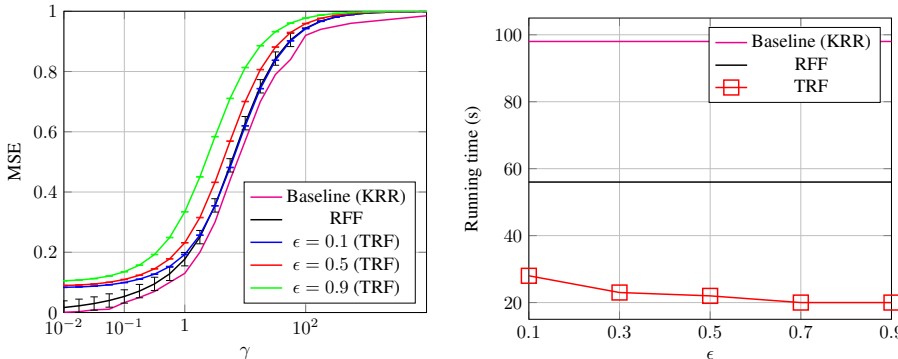

Figure 5: Testing MSE of kernel ridge regression as a function of regularization parameter $\gamma$, $p = 512, n = 1024, n_{test} = 512, m = 512$. Ternary function (with thresholds $s_-, s_+$ chosen to match gaussian moments of $[\cos, \sin]$ function) with $\mathbf{W}$ distributed according to equation 4 with $\epsilon \in \{0.1, 0.3, 0.5, 0.7, 0.9\}$ versus KRR and RFF. GMM dataset with $\boldsymbol{\mu}_a = [\mathbf{0}_{a-1}; 4; \mathbf{0}_{p-a}], \mathbf{C}_a = (1 + 4(a - 1)/\sqrt{p})\mathbf{I}_p, p = 512, n = 2048$. Results averaged over 5 independent runs.

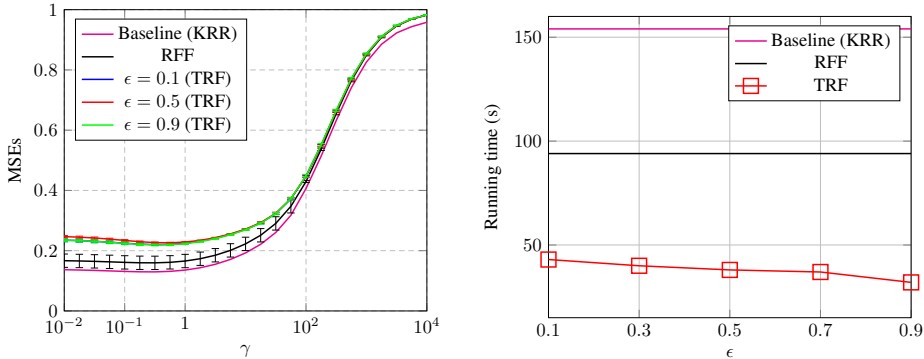

Figure 6: Testing MSE of kernel ridge regression as a function of regularization parameter $\gamma$, $p = 512, n = 1024, n_{test} = 512, m = 4096$. Ternary function (with thresholds $s_-, s_+$ chosen to match gaussian moments of $[\cos, \sin]$ function) with $\mathbf{W}$ distributed according to equation 4 with $\epsilon \in \{0.1, 0.3, 0.5, 0.7, 0.9\}$ versus KRR and RFF. GMM dataset with $\boldsymbol{\mu}_a = [\mathbf{0}_{a-1}; 4; \mathbf{0}_{p-a}], \mathbf{C}_a = (1 + 4(a - 1)/\sqrt{p})\mathbf{I}_p, p = 512, n = 2048$. Results averaged over 5 independent runs.

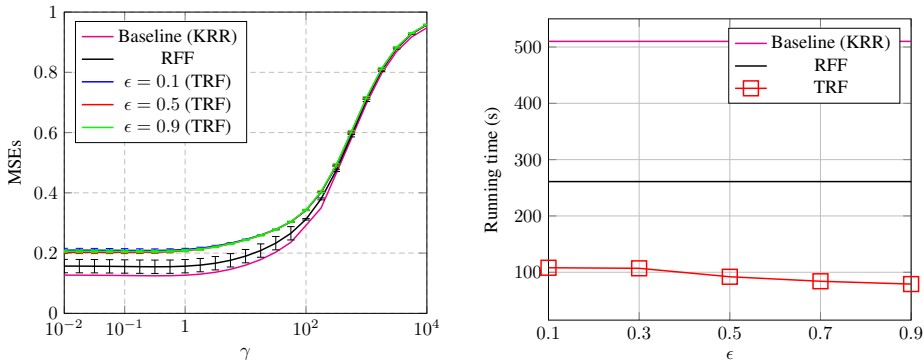

Figure 7: Testing MSE of kernel ridge regression as a function of regularization parameter $\gamma$, $p = 512, n = 1024, n_{test} = 512, m = 10^4$. Ternary function (with thresholds $s_-, s_+$ chosen to match gaussian moments of $[\cos, \sin]$ function) with $\mathbf{W}$ distributed according to equation 4 with $\epsilon \in \{0.1, 0.3, 0.5, 0.7, 0.9\}$ versus KRR baseline and RFF. GMM dataset with $\boldsymbol{\mu}_a = [\mathbf{0}_{a-1}; 4; \mathbf{0}_{p-a}], \mathbf{C}_a = (1 + 4(a - 1)/\sqrt{p})\mathbf{I}_p, p = 512, n = 2048.)$. Results averaged over 5 independent runs.

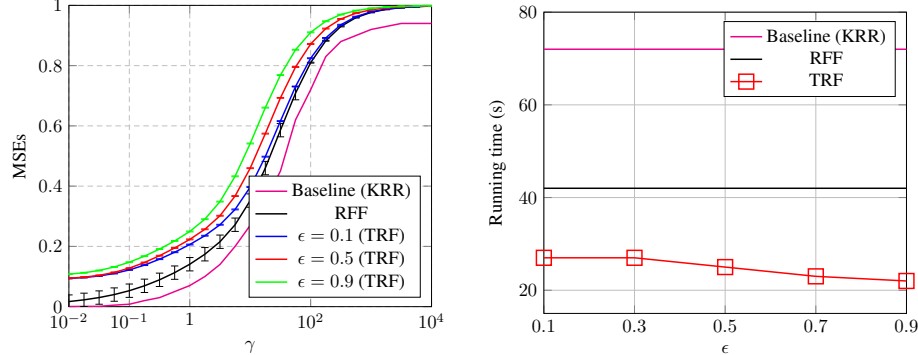

Figure 8: Testing mean squared errors (MSEs, **LEFT**) and running time (**RIGHT**) of kernel ridge regression as a function of regularization parameter $\gamma$, $p = 512, n = 1024, n_{test} = 512, m = 512$. Ternary function (with thresholds $s_-, s_+$ chosen to match the Gaussian moments $d_1, d_2$ of $[\cos, \ \sin]$ function) with $\mathbf{W}$ distributed according to (4) with $\epsilon \in \{0.1, 0.3, 0.5, 0.7, 0.9\}$, versus KRR and RFFs on MNIST dataset 2 classes - digits $(7, 9)$. Results averaged over 5 independent runs.

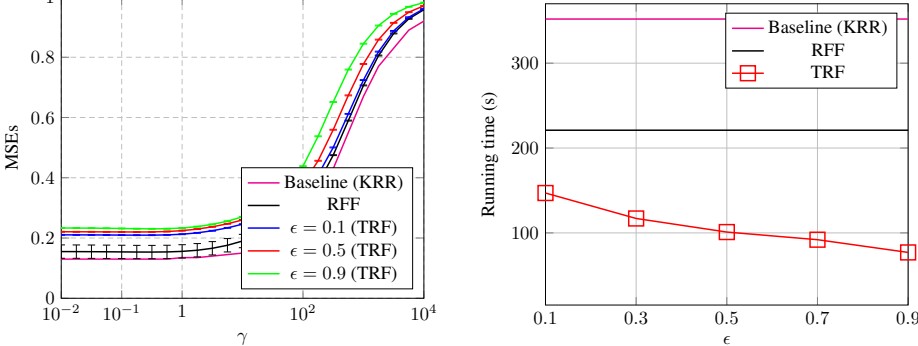

Figure 9: Testing mean squared errors (MSEs, **LEFT**) and running time (**RIGHT**) of kernel ridge regression as a function of regularization parameter $\gamma$, $p = 512, n = 1024, n_{test} = 512, m = 10^4$. Ternary function (with thresholds $s_-, s_+$ chosen to match the Gaussian moments $d_1, d_2$ of $[\cos, \ \sin]$ function) with $\mathbf{W}$ distributed according to (4) with $\epsilon \in \{0.1, 0.3, 0.5, 0.7, 0.9\}$, versus KRR and RFFs on MNIST dataset 2 classes - digits $(7, 9)$. Results averaged over 5 independent runs.

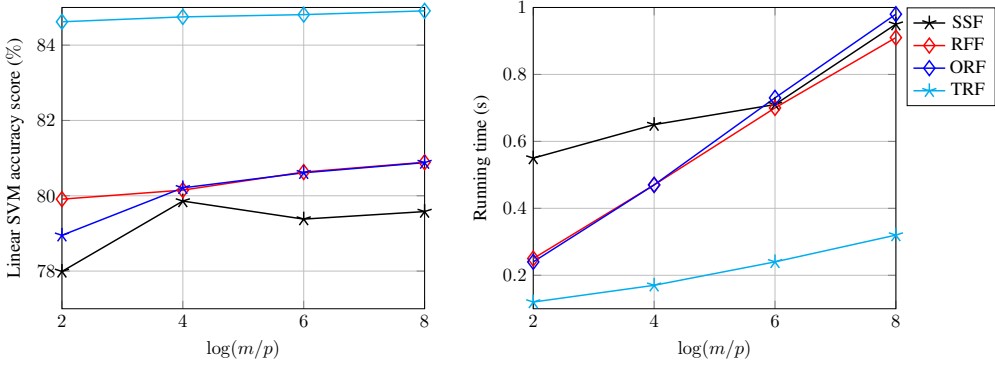

Figure 10: Test accuracy using libsvm on different state-of-the-art random features kernels. a8a (UCI) dataset. Number of training samples $n = 22696$ - number of test samples $n_t = 9865$, varying ratio $\log m/p$ number of random features over dimension $p = 123$. Note that the y-axis is zoomed in to better distinguish the performance of different methods.

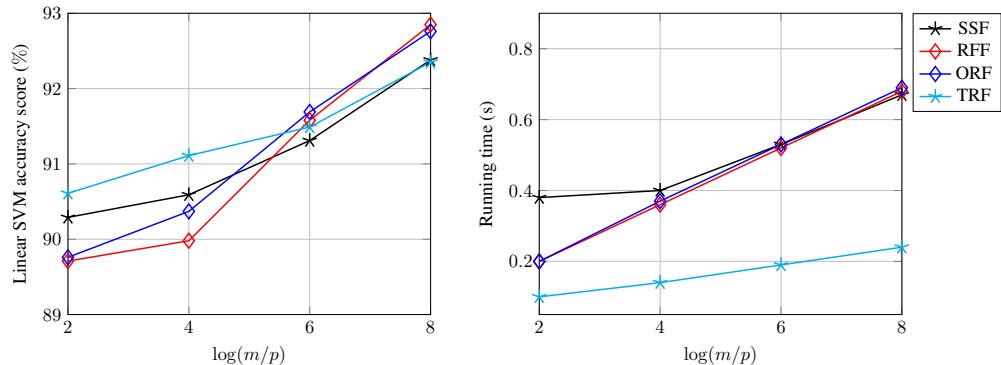

Figure 11: Test accuracy using libsvm on different state-of-the-art random features kernels. IJCNN1 (Prokhorov, 2001) dataset. Number of training samples $n = 49990$ - number of test samples $n_t = 91701$, varying ratio $\log m/p$ number of random features over dimension $p = 22$. Note that the y-axis is zoomed in to better distinguish the performance of different methods.

**Random features based Support Vector Machine** We empirically evaluate the classification performance of various random features approximation algorithms, on several benchmark datasets. We compared the different algorithms (RFF (Rahimi & Recht, 2008), ORF (Yu et al., 2016), SSF (Lyu, 2017)) on 3 datasets IJCNN1 (Prokhorov, 2001), Cov-Type, and a8a from the UCI ML repository considered in (Liu et al., 2021a), with our TRF method where we choose the thresholds coefficients $s_-, s_+$ according to Algorithm 1 (to match the generalized Gaussian moments of Gaussian kernel). Figure 10 shows the results for the a8a dataset, Figure 11 for the IJCNN1 dataset and Figure 12 for the Cov-Type dataset. The lower running time along with higher SVM test accuracy indicates the superiority of our RMT-inspired TRF method over the other Gaussian kernel approximation methods.

**Comparison of TRF with equivalent $0^{th}$ order Arc-cosine kernel (with ReLU function)** We consider Support Vector Machine (SVM) classification with random features Gram matrix $\mathbf{G}$ on Fashion MNIST data (LeCun et al., 1998) and on VGG16 embeddings of CIFAR10 (Krizhevsky et al., 2009) and Imagenet (Deng et al., 2009) datasets in Figures 14 -13 -15 respectively. We use VGG16 with batch normalization (Ioffe & Szegedy, 2015) pre-trained on ImageNet (Deng et al., 2009) as a feature extractor. We fine-tune this model on the CIFAR10 dataset with 240 epochs and a mini-batch size 64 with a SGD optimizer with momentum 0.9 and an initial learning rate of 0.1. We then extract the output of the first fully connected layer of the classifier as our features. For Imagenet, we directly extract the features from the pretrained model. We compare (i) RF with $\sigma(t) = \max(t, 0)$ (ReLU) and Gaussian $W_{ij} \sim \mathcal{N}(0, 1)$ (known as the 0th order Arc-cosine kernel) to (ii) the proposed TRF

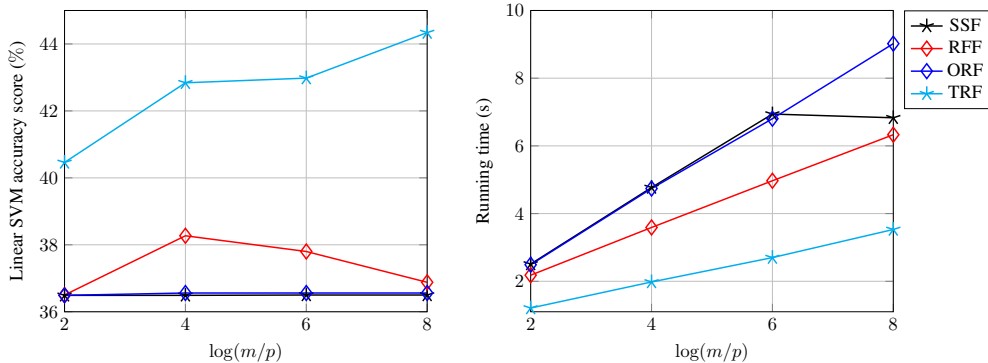

Figure 12: Test accuracy using libsvm on different state-of-the-art random features kernels. Cov-Type dataset. Number of training samples $n = 49990$ - number of test samples $n_t = 91701$, varying ratio $\log m/p$ number of random features over dimension $p = 22$.

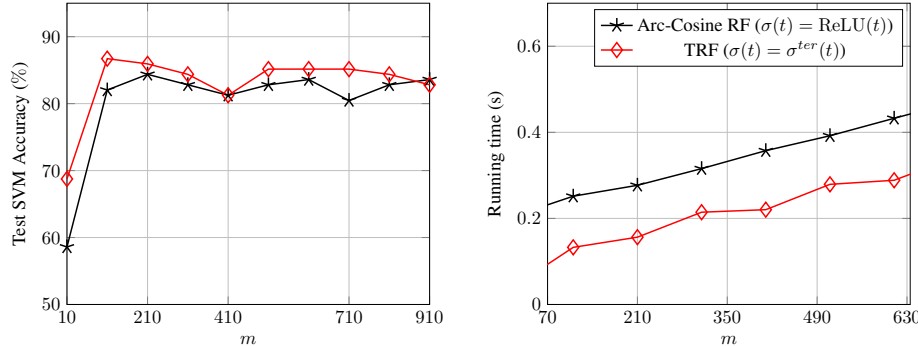

Figure 13: SVM test accuracy using different kernels. VGG-16 Embeddings of CIFAR10 dataset (Number of features $p = 4096$). Number of samples $n = 1024$ fixed, varying number of random features from $m = 10$ to $m = 1800$.

method with $\sigma^{ter}(t)$ in (3) and ternary random projection matrix $\mathbf{W}^{ter}$ defined in (4). The thresholds $s_-, s_+$ of $\sigma^{ter}$ are tuned in such away that the generalized Gaussian moments $d_1$ and $d_2$ are matched with those of ReLU (see Table 1), as described in Corollary 1 and Algorithm 1. Figures 14 -13 -15 display the test SVM accuracy as a function of the number of random features, for our TRF method compared to the random features corresponding to the Arc-Cosine kernel. We observe similar test performances with the two kernels while having a computation and storage gains for the ternary kernel.

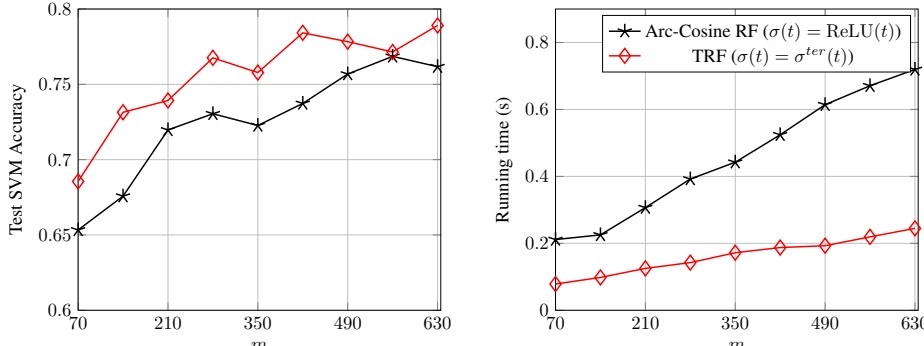

Figure 14: SVM test accuracy using different kernels. Raw Fashion-MNIST dataset (Number of features $p = 784$). Number of samples $n = 1024$ fixed, varying number of random features from $m = 70$ to $m = 700$. Note that the y-axis is zoomed in to better distinguish the performance of different methods.

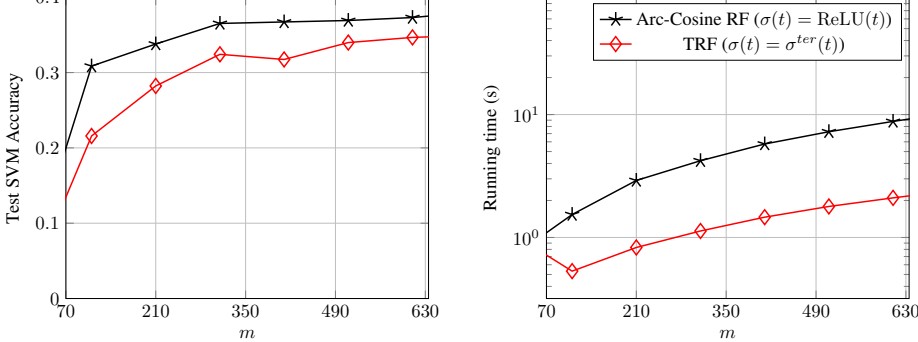

Figure 15: SVM test accuracy using different kernels. LeNet-64 Embeddings of Imagenet dataset (Number of features $p = 2018$). Number of samples $n = 1024$ fixed, varying number of random features from $m = 10$ to $m = 1800$.

