# OpenReview forum: "Random matrices in service of ML footprint: ternary random features with no performance loss"
_ICLR.cc/2022/Conference — ICLR 2022 Poster_

### Official Review · Reviewer_c4jW · 2021-11-02

**Correctness:** 3
**Technical Novelty And Significance:** 2
**Empirical Novelty And Significance:** 2
**Recommendation:** 6
**Confidence:** 2

**Main Review:**

This paper is well organized. I like the idea of the paper to exploit the theoretic result (in the high dimensional limit the spectrum of the kernel only depends on the first two generalized Gaussian moments of the activation function) by coming up with a sparse choice of the activation function and projection weight's distribution, that could make the random features computationally efficiently and easier to store.

The main concern is the importance of this finding (have a constant scale improvement in computation time and storage memory). I believe the proof technique of the main theorem is not original, so the contribution mainly come from the gains in computation and storage. I feel the authors should spend more space to argue that.

Some minor issues:
- In page 7 the authors argue that, the computation of the proposed random feature requires no multiplication and only $O(\epsilon mnp)$ additions. Could there be more explanations around this claim? For example, (1) in the case $\epsilon != 0$, why there is no multiplication needed, (2) in the case $\epsilon = 0$, does the proposed random feature requires no multiplication and no additions? Or should the needed additions be $O((1-\epsilon) mnp)$ instead since $1-\epsilon$ proportion of the weights are non-zero?

-------------- After revision ---------------

Thanks the authors' response. After reading the response and other reviewers' comments, I agree that the constant scale improvement in computation time and storage is a good contribution, and I raise my score to 6.

**Summary Of The Paper:**

This paper studies random features method, which randomly projects the data onto a low-dimensional space, while at the mean time approximates the original kernel structure.
Some contributions of this paper:
- The authors show that, under the Gaussian mixture model, in the high-dimensional limit, in the spectrum sense, the random features-type kernel (defined in equation (1) of the paper) only depends on the first two generalized Gaussian moments of the activation function (with the assumption that the random projection weights are i.i.d. from some mean 0 and var 1 distribution).
- Based on the above result, the authors propose a special choice of the random weights and the activation function, that could be efficiently computed and needs less memory to store.
- The authors provide empirical evidence showing that the proposed method performs as well as other random features methods, and with advantage in computation and storage.

**Summary Of The Review:**

My biggest concern is how important the main contribution is (have a constant scale improvement in computation time and storage memory). To be honest I'm not sure how to measure that, and thus will lower my confidence score.

---

> ### Author Response · Authors · 2021-11-18
> **Response to Reviewer c4jW**
>
> We would first like to thank the reviewer for the time reviewing our work and providing helpful remarks as long as identifying some typos. In the following, we provide a step-by-step response to the comments of the reviewer.
>
> * **on the importance of the theoretical contribution**:
> 	From a theoretical perspective, the proposed analysis is novel: while clearly built upon previous efforts (e.g., the ideas and some proof techniques in (Fan and Wang, 2020) and (Liao and Couillet, 2018b)), our results improve previous efforts such as Theorem 1 in (Liao and Couillet, 2018b) by considering a much large family of random matrices $W$ and nonlinear function $\sigma$, see Footnote 4 for a detailed discussion on this point.
>
> 	From a practical perspective, we proposed, based on the theoretical analysis, the TRF approach that, as has been shown in the extensive experiments in Section 4 and Section A.6 of the appendix, outperforms SoTA random features compression/quantization methods on many popular real-world datasets.
>
> * **some minor issues**: we thank the reviewer for spotting this typo. The number of additions required is indeed $O((1-\epsilon)mnp)$ instead of $O(\epsilon mnp)$; this was a typo and has been fixed in the revised version. However, independently of the sparsity, there is no multiplications involved as we are multiplying vectors with entries (-1, 1, 0) with a real entries vector; so we just need to identify the position of the positive, negative and null entries and perform additions.

---

### Official Review · Reviewer_RdKc · 2021-11-02

**Correctness:** 3
**Technical Novelty And Significance:** 4
**Empirical Novelty And Significance:** 3
**Recommendation:** 6
**Confidence:** 3

**Main Review:**

Strengths:
1)	This paper provided self-contained theoretical guarantees for the asymptotic equivalent of the primal kernel and a delicate form of ternary random features. The asymptotic results are interesting, and the proofs seems to be correct.
2)	This paper designed a simple and efficient framework to construct ternary random features. Due to the sparsity of ternary random features, the algorithm is benefit from both high computational efficiency and lower storage complexity.
3)	This paper also provided experimental validations by comparing the training time, memory costs and MSE with SOTA methods. The empirical results coincided with the theoretical finding, that the proposed approach characterized both computational and storage gains.
4)	The writing of this paper is clear, and most parts are easy to follow.

Weakness:
1)	My main concern is that the Gaussian mixture data assumption may be too strong. As shown in Eq. (7), this paper assumed the inputs are under multivariable Gaussian distribution. Many real-world datasets break this condition, such as long-tailed distribution. Moreover, the classic random features, for example random Fourier features, have no restriction on data distribution. Besides the Gaussian data assumption, Assumption 1 (iii) also seems to be strict. The authors are expected to provide more examples to illustrate the applicability of these assumptions.
2)	The presented asymptotic theory requires the dimension of input space to approach infinity $p \to \infty$ is unfamiliar in practical. Because the dimension of input space is fixed, I wonder that is there still a good approximation between $K$ and $\tilde K$ if $p$ is small? For a given task with a fixed $p$, is there a natural gap between $K$ and $\tilde K$?
3)	It seems both Theorem 1 and Corollary 1 are independent from the required number of random features $m$. In the existing random features literature, $m$ is crucial to the approximation ability and generalization ability. In general case, $m=O(\sqrt{n})$ random features can guarantee the similar generalization ability (Rudi and Rosasco, 2017). The authors may illustrate how the number of ternary random features influence the approximation or generalization.
4)	The kernel hyperparameters usually determine the performance of kernel methods, but the proposed random features approach seems to be independent from kernel hyperparameters and only depend on the kernel type. Can TRF approximate any kernel with different hyperparameters? How does TRF remove the influence of kernel hyperparameters?

Rudi A, Rosasco L. Generalization Properties of Learning with Random Features[C]//NIPS. 2017: 3215-3225.


**Summary Of The Paper:**

With Gaussian mixture assumption for input data, this paper proposed a sparse random features approach where the approximated kernel is independent of the iid weights and depends on Gaussian moments of the activation functions. This paper provided a theoretical guarantee for the asymptotical equivalence with the centered kernel. And then, the authors derived the computationally efficient random features and devised a simple algorithm. Finally, they validate the accuracy and efficiency of the proposed random features by several experiments.

**Summary Of The Review:**

This paper provided the sparse random features with theoretical guarantees, efficient algorithm and sufficient experimental validations. However, this paper assumed the input data with Gaussian mixture that may limit its applicability. Meanwhile, the influence of the number of random features has not been explored well.

---

> ### Author Response · Authors · 2021-11-18
> **Response to Reviewer RdKc**
>
> We thank the reviewer for for the time reviewing our work and providing helpful remarks and raising questions that need clarification. In the following, we provide a step-by-step response to the comments of the reviewer.
>
> * **on the Gaussian mixture data assumption**:
> 	The main reason for the use of Gaussian Mixture Models is for the mathematical tractability of the random features kernel matrix as a function of the data first and second-order statistics. To the best of our knowledge, there is no such explicit data-statistics-dependent characterization of random features kernels in the literature.
>
> 	From a theoretical perspective, our results, as stated in Remark 1 of the paper and with some additional efforts, extend to a much broader family of data distributions (in fact, to the large family of concentrated random vectors). It has been shown in (Seddik et al, 2020) that artificial images generated by a GAN (that look extremely close to real images), by definition/construction, belong to this family. More importantly, (Seddik et al, 2020) showed that in the large dimensional regime (as in our Assumption 1), the performance of kernel methods on any concentrated random vectors asymptotically only depend on the first- and second-order statistics of the distribution and thus match the performance on a simple GMM. As a result, we believe it is a theoretically reasonable choice to start with GMM data to investigate the large dimensional behavior of random feature methods.
>
> 	This "large dimensional universality" argument is also empirically supported by the extensive experiments (in Section 4 and Section A.6 in the appendix) on *real-world* datasets, where the proposed TRF approach is observed to perform well across a wide range of real-world datasets and problems.
>
> * **on the large dimension $p$**: for comparably large $n,p$, the difference in operator norm $\| K - \tilde{K} \|$ decays like $1/sqrt{p}$ with high probability: in a sense, we do not practically need an increasing data dimension $p$ for Theorem 1 to hold, but only that "the dimension $p$ is fixed and numerically large" so that the approximation error is small. In the experiments, we observe that the approximations are relatively accurate for the dimension of natural images (see for example Figure 3 on MNIST data with dimension $p=784$).
>
> * **on the number of random features $m$**: we would like to emphasize on the fact that our main theoretical results (in Theorem 1 and Corollary 1) are on the spectral behavior of the expected/limiting random feature kernel obtained by taking the number of random features $m \to \infty$: further control on the difference between the empirical and expected kernel $\| G - K \|$ (for random features Gram matrix $G$ defined in (6)) can be obtained with a (matrix) concentration argument. We have added Remark 4 to clarify this point (in Section A.5 of the appendix for the moment due to space limitation, and will be moved back to the main text in the final version of the manuscript).
>
> * **on the kernel hyperparameters**: note that in the case of RFF, which approximates Gaussian kernel of the type $\exp(\| x_i - x_j \|^2/ (2 \sigma^2))$, it suffices to draw the entries of $W$ from $\mathcal{N}(0, 1/\sigma^2)$. The obtained random feature matrix is the same as rescaling all data points by $1/\sigma$ and drawing $W$ from the standard Gaussian distribution $\mathcal{N}(0,1)$. In this sense, TRF does not remove the influence of the hyperparameters, but it is devised for kernels having unit scale parameters, and can be easily extended to the case of non-unit (but finite) scale parameters.

---

### Official Review · Reviewer_AjvY · 2021-11-02

**Correctness:** 4
**Technical Novelty And Significance:** 3
**Empirical Novelty And Significance:** 4
**Recommendation:** 8
**Confidence:** 3

**Details Of Ethics Concerns:**

No concern.

**Main Review:**

# Quality and Clarity

The paper is well written and clear. It was easy to follow. I'm not a fan of purely kernel oriented papers discussing deep learning in the introduction if that's not significant to the paper, but that's a personal subjective preference. Good marks for quality and clarity

# Theory

The core theoretical claim is compelling in its generality and formality. While it wouldn't be enough to carry the paper on its own, it serves as a nice motivation for the TRF algorithm proposed. There's really only one theoretical claim in the paper, but that claim is all they need to show. It's well stated, clear, and to the point. Elegant.

Amongst the theory, the only clarification I would like to see is why the estimator $\hat\tau = \frac1n \sum_{i=1}^n \|x_i\|^2$ is asymptotically correct. I assume this follows from some simple properties of the Gaussian distribution, but it would be nice to see at least this discussed in the paragraph before _Algorithm 1_.

As a side note: while I didn't review the full proof of Theorem 1, glancing at the appendix, the proof seems surprisingly (in a good way) simple and approachable for a RMT proof. Props.

# Experiments

The theory in this paper serves as a motivation for the TRF algorithm which approximate the kernel matrix. Unlike most work on random features, this paper's theory requires data to be drawn from a gaussian mixture model (GMM). Since most theory of kernel approximations don't require this assumption, in my view, this paper lives and dies by its experiments: is the GMM essential to the proposed approach, or is it a mathematical simplification to make the math more tractable and motivation a pragmatic algorithm?

The experiments are largely convincing. There's several points I would very much like to see (frankly I expect to see) fixed up in a camera-ready version. But overall, the experiments are compelling, so I think the overall paper works well. This suggests the GMM assumption in the theory was just a simplification to make the math tractable.

Overall, the experiments compare three statistics: running time, space complexity, and the statistical performance of the resulting kernel matrix. The proposed TRF algorithm is compared usually against classical Random Fourier Features (RFF), and sometimes against a few other kernel approximation algorithms.

The experiments suggest that TRF is notably faster and uses less space than RFF. Further, they can tune a sparsity parameter to tradeoff the accuracy of the kernel matrix with the space of the kernel matrix. Some interesting takeaways from these experiments:
- A very sparse kernel sometimes does not impact accuracy a lot, and sometimes is very important (figures 5, 6, and 7 show sparsity control accuracy for only moderate amounts of regularization). Is there a rational about _when_ sparsity impacts accuracy a lot?
- Sparsity helps speed up computation, but not a huge amount. Going from a $90\%$ nonzero matrix to a $10\%$ nonzero matrix can shave off $\frac13$ of the computation time. It's a good boost, but maybe bad value if this can increase error by $50\%$?
    - The $50\%$ number comes from Figure 5, $\gamma\approx5$, where the triangle has MSE $0.5$ and the $\otimes$ has error $0.75$.
- The running time improvements are especially encouraging across the board.

These experiments are strong enough (suggesting that the theory -- as they conjecture -- is more general than the GMM setting) that I support clearly accepting the paper.

That said, here are the changes I would like/expect to see in an experiment driven paper like this:
1. Better baselines. We see most plots use classical RFF as a baseline, but this doesn't always make sense. In Figure 3, we see the $\varepsilon=0.1$ data fall below the baseline MSE, suggesting that TRF outperforms RFF. That makes for a weird baseline. I would like to see the true optimal MSE, computed without any random features at all. I would like to see this on all the statistical efficiency (MSE and accuracy) plots.
    - This will add a good deal more of valuable information: like in Figure 3 on the left plot, we see a gap between all TRF models and the RFF model. Is the RFF model close to the true baseline, or do TRF and RFF both have a large gap from the baseline? I understand there is some approximation error, but I don't understand if there's a lot of approximation error.
1. Confidence intervals. TRF and RFF are both randomized algorithms. RFF has many well studied concentration inequalities, so I'm reasonably confident it's well concentrated. I don't know that about TRF though. What do the $25^{th}$ and $75^{th}$ quantiles look like? Also, how many repetitions did you run these algorithms for?
1. For Figures 5, 6, and 7, where there's a lot of series of data, these plots are too hard to read. I would recommend reducing the number of series (i.e. drop $\varepsilon=0.3,0.7$) to make the lines more legible when the confidence intervals are added, and to color code the series instead of using different markers.

---
# Extra Tidbits

## Some Technical Questions I Have
1. In the experiments, you mention using 32 bit floats for Nystrom approximation. To reduce space complexity, can you just perform the same computations with 16 bit floats or 8 bit floats. Does this have a disastrous statistical impact, or could we close the gap between TRF and Nystrom a bit, very easily?
1. Are you aware of Michael Mahoney's work on Random Matrix Theory for deep learning? I read one or two of those papers a while back, and they might also fit into this world of RMT to understand efficient estimators (e.g. in the end of your Section 1.3)? Not totally sure, and I don't want to force you to add in a reference to an unrelated paper. Just want to bring up a possibly interesting connection existing in the literature.
1. When does TRF give better statistical performance than RFF? (Figure 3 left image, large $\gamma$ and small $\varepsilon$)

## Typos and small recommended edits
1. [Page 2 footnote] "is equivalent to _centering_ the data" not "center"
1. [Figure 4] Replace "$0.5 \cdot 10^{-2}$" with "$0.05$"
1. [Figures 11-15] Mention in the captions that the y-scale is zoomed in a lot. Explicit is better than implict.
1. [Figure 12] Add the dataset name to the caption. I know it's mentioned in the text of the appendix, but it should be here too.


**Summary Of The Paper:**

The paper proposes a way to approximate common kernel matrices in a way that is (1) Sparse (2) Low Bit-Complexity (i.e. the nonzero entries have few bits) (3) efficient to compute.

The theoretical justification from this model comes from a random matrix theory (RMT) analysis of kernel matrices for datasets of vectors drawn iid from a fairly general gaussian mixture model. The RMT analysis shows that the true kernel matrix is asymptotically distributed as the approximation described above. No non-asymptotic results are given.

The introduction mentions deep networks, but this paper really has no strong connection to deep learning. View it as a kernel paper.

A good amount of supporting experiments are given to demonstrate that the approximate kernel matrix has the same statistical performance as (eg) a Random Fourier Features approximation.

**Summary Of The Review:**

The theory is a compelling motivation for a proposed matrix approximation algorithm that seems to experimentally work well.

They should clean up their experiments a little, but this is overall encouraging as a pragmatic way to reduce the time and space complexity of real kernel algorithms by large constant factors.

I recommend this paper.

---

> ### Author Response · Authors · 2021-11-18
> **Response to Reviewer AjvY**
>
> We would first like to thank the reviewer for the time reviewing our work and providing helpful and detailed  remarks. This has helped in improving the quality of the paper especially in the experimental part. In the following, we provide a step-by-step response to the comments of the reviewer.
>
> * **estimation of the key parameter $\tau$**: We have added a sentence to explain the line of arguments for this estimate before Algorithm 1, the detailed proof of which is placed in Lemma 1 of the appendix.
>
> * **impact of sparsity on accuracy**: note that our theoretical results technically hold only in the (relatively) "dense" regime, that is, then the number of nonzero entries of $W \in \mathbb{R}^{m \times p}$ is of order $O(mp)$, we conjecture this no longer holds if $W$ is more sparse (e.g., having nonzero entries of order $o(mp)$) and the sparsity will have a much larger impact on the accuracy in that regime.
>
> * **on the experiments**: we have re-run all random features-based ridge regression experiments, by plotting the $\pm 1$ standard deviation of the performances (obtained over 5 different realizations). In particular, We have added as baseline the performance of kernel ridge regression (KRR) by using the corresponding expected/limiting kernel without any random features (or equivalently, by letting the number of random features $m \to \infty$). As expected (see, e.g., Figures 3, 5-8), the performances of RFF and TRF are  close to the KRR baseline when the number of random features $m$ is large. Also, we note from these figures that the standard deviations of TRF are significantly smaller than those of RFF, which appears as another empirical advantage of the proposed TRF approach.
>
> * **compared to Nystrom with 16 bits**: note that the classical Nystrom approximation is a full precision method. To perform computations with 16 bits or 8 bits floating number, we need a "quantized Nystrom" method which, to the best of our knowledge, is not yet present in the literature. As a consequence, in Figure 1 and 4, we compared the original Nystrom (32 bits) and our TRF to the LP-RFF (1 and 8 bits) which is a quantization approach for RFF.
>
> * **When does TRF give better statistical performance than RFF?**:
> Theoretically speaking, when the number of random features $m \to \infty$ (in fact $m/max(n,p) \to \infty$ as in Remark 4 of the appendix, so that the empirical Gram matrix converges to the expected limiting kernel), TRF should give the same statistical performance as RFF. In the left plot of Figure 3, the performance gap between TRF and RFF is due to the finite-dimensional effect (of the number of random features), which can be closed by increasing the number of random features $m$. Establishing non-asymptotic results that hold for any value of $n,p,m$, while providing a more precise theoretical understanding on the statistical performance between TRF and RFF, is out of the scope of this work.
>
> * *Minor comments*:
> Thanks for the recommended edits. They are all included in the revised version.

---

> > ### Comment · Reviewer_AjvY · 2021-11-29
> > **Still happy with a score of 8**
> >
> > Thanks to the authors for the response. I find the plots vastly easier to read an interpret, and I agree that the low variance of TRF is also very encouraging. There's a small point I'd like to touch on with the authors, but overall, I'm happy maintaining a confident score of 8 for this paper.
> >
> > **This paper should clearly be accepted.**
> >
> > ---
> >
> > Nystrom with floating point numbers is already a quantized Nystrom, right? Floating point numbers are quantized real numbers. Can't we just naively perform the same computation as the 32 bit floats, but on smaller 16 floats?
> >
> > _(sorry I didn't post this sooner, no worries if you can't respond to this last question)_

---

> > > ### Author Response · Authors · 2021-12-01
> > > **Regarding Nystrom quantization**
> > >
> > > We thank the reviewer for clarification on the Nystrom method. Viewing floating point as quantization of real numbers, we can then run the Nystrom method with 16 floats and consider it as Quantized Nystrom 16 bits. We will add that in the final version of the paper.

---

### Official Review · Reviewer_2gn7 · 2021-11-03

**Correctness:** 3
**Technical Novelty And Significance:** 4
**Empirical Novelty And Significance:** 4
**Recommendation:** 8
**Confidence:** 2

**Main Review:**

I like the ternary quantization idea and it seems to work great in the practical experiments. It seems to give roughly a 2x speedup and 8x memory decrease compared to RFF and other kernel methods, which is quite impressive. The theoretical results are interesting but they work under quite restrictive assumptions.

I have concerns on the quality of presentation of the theoretical results. In particular, I am sure the statement of Theorem 1 can be made clearer. For example, the quantities defined after (8) seem to be in somewhat random order which makes it hard to read. Also, it would be good to flesh out what is the main *technical* insight of Theorem 1 compared to previous work. More importantly, I was not able to follow the proof of Theorem 1. My suggestion is to add more text to explain what is happening and re-write equations like (19) in a way that is easier to parse. Also, I don't see how Corollary 1 follows from Theorem 1. E.g. where is (10) coming from? but also there is a lot of explanation missing here since the approximate kernels in Theorem 1 and Corollary 1 are very different.

I would like to see a discussion about the assumption that the entries of $w$ are i.i.d. In particular, when (and why) should I expect this assumption to hold?

I didn't see any discussion on the minimum number of random features $m$ needed. Is this not necessary in the analysis?

What is the time to compute the random features? It seems like it has some non-trivial parts like computing $\hat{s}^+$, $\hat{s}^{-}$, so it would be good to have a short discussion on this (with some sample runtimes), and also explain a bit more how $\hat{s}^{+}, \hat{s}^{-}$ are computed.

**Summary Of The Paper:**

The authors present a resource-efficient approach for approximating kernels using random feature transformations. The random feature approach was introduced in a seminal paper by Rahimi and Recht, where each random feature is extracted from the input $x$ as $\sigma(\langle w, x\rangle )$, for a random (appropriately distributed) vector $w$ and an activation function (e.g. sinusiod) $\sigma$. In this paper, the goal is to save memory and time by restricting $w$ and $\sigma$ to take values in {$-1,0,1$}, meaning that $\sigma(\langle w, x\rangle )$:
1. can be evaluated only with additions/subtractions (no multiplications) and
2. only takes $1$ bit of storage.

The authors prove that under some conditions, their kernel estimate approaches the true kernel at the limit where the number of features and the number of data points are comparable and approach infinity. The assumptions are roughly:
1. The data follows a Gaussian mixture model,
2. the input features are approximately pairwise orthogonal,
3. the entries of $w$ are chosen i.i.d. and have bounded fourth moment, and
4. some boundedness assumptions on the generalized Gaussian moments of $\sigma$.

The theoretical result is accompanied by numerical experiments on various tasks like kernel ridge regression and SVM on both real and synthetic datasets. The results demonstrate a significant advantage of the authors' approach compared to previous quantized random features approches, both in terms of runtime and memory, without a significant drop in accuracy.

**Summary Of The Review:**

In summary, this paper is mostly well-written except for the theoretical parts which need some work. The idea is really interesting and relevant for resource-intensive applications. The experimental results look very promising. I am leaning towards acceptance, but I might change my rating based on if my concerns have been resolved.

---

> ### Author Response · Authors · 2021-11-18
> **Response to Reviewer 2gn7**
>
> We would first like to thank the reviewer for for the time reviewing our work and providing helpful remarks. In the following, we provide a step-by-step response to the comments of the reviewer.
>
> * **presentation of theoretical results**: we thank the reviewer for pointing this out to improve the readability of this article. We have added explanations for the technical insight of Theorem 1 and re-organized the presentation of Theorem 1 as well as its proof, please see the texts marked in red in Section A.3 of the appendix. We have also added clarifications and detailed derivations for Corollary 1, see Section A.4 in the appendix. As for the comment on the fact that the quantities defined after (8) are seemingly randomly organized, we consider that it helps readability when defining terms that have similar significance together. For example, the class statistics (M, t, T), the random vectors (Z, \phi).
>
> * **i.i.d. assumption of $W$**: We have added a discussion in Remark 2 of Section A.1 in the appendix, and provided examples of popular random features approaches for which this i.i.d. assumption holds (e.g., for vanilla random Fourier features) and fails to hold (e.g., for data dependent RF approaches such as leverage score based methods). In particular, it was shown in (Rahimi and Recht, 2008) that the assumption holds for the Random Fourier Features approximating the Gaussian kernel with unit bandwidth. For general bandwidth hyper-parameters, it suffices to rescale the data and our technical results remain valid.
>
> * **on the number of random features $m$**: we would like to emphasize on the fact that our main theoretical results (in Theorem 1 and Corollary 1) are on the spectral behavior of the *expected/limiting* random feature kernel obtained by taking the number of random features $m \to \infty$, further control on the difference between the empirical and expected kernel $\| G - K \|$ (for random features Gram matrix $G$ defined in (6)) can be obtained with a (matrix) concentration argument. We have added one sentence after Theorem 1 and Remark 4 to clarify this point (in Section A.6 of the appendix for the moment due to space limitation, and will be moved back to the main text in the final version of the manuscript).
>
> * **computing of the parameters $\hat s_-$ and $\hat s_+$**: We now elaborate before Algorithm 1, to explain in more detail on how $\hat s_-$ and $\hat s_+$ are computed. As $\hat s_-$ and $\hat s_+$ are solutions of fixed point equations that do not depend on the problem dimensions, the computational complexity is $O(1)$, and in practice this system of nonlinear equation can be solved in a few seconds.

---

### Decision · Program_Chairs · 2022-01-20

**Decision:**

Accept (Poster)

**Comment:**

The reviewers overall were quite happy after the rebuttal phase, in which the authors considerably improved the presentation quality and addressed reviewer concerns, and recommended acceptance. The reviewers agreed that while the theory was short and relied on various possibly restrictive assumptions and maybe was largely an improvement in constant factors, it extended prior work (some of which was in ICLR) and was interesting and motivated the experiments which were notably faster than existing methods.